**ARTICLES**

# Expert-level detection of pathologies from unannotated chest X-ray images via self-supervised learning

Ekin Tiu[1,2,4], Ellie Talius[1,2,4], Pujan Patel[1,2,4], Curtis P. Langlotz[3], Andrew Y. Ng[1] and Pranav Rajpurkar [2 ✉]

In tasks involving the interpretation of medical images, suitably trained machine-learning models often exceed the performance of medical experts. Yet such a high-level of performance typically requires that the models be trained with relevant datasets that have been painstakingly annotated by experts. Here we show that a self-supervised model trained on chest X-ray images that lack explicit annotations performs pathology-classification tasks with accuracies comparable to those of radiologists. On an external validation dataset of chest X-rays, the self-supervised model outperformed a fully supervised model in the detection of three pathologies (out of eight), and the performance generalized to pathologies that were not explicitly annotated for model training, to multiple image-interpretation tasks and to datasets from multiple institutions.

Deep learning has enabled the automation of complex medical image interpretation tasks, such as disease diagnosis, often matching or exceeding the performance of medical experts[1–5]. However, despite these meaningful improvements in diagnostic efficiency, automated deep learning models often require large labelled datasets during training[6]. These large-scale labelling efforts can be expensive and time consuming, often requiring extensive domain knowledge or technical expertise to implement for a particular medical task[7,8].

Several approaches such as model pre-training and self-supervision have been proposed to decrease model reliance on large labelled datasets[9–12]. Although self-supervised pre-training approaches have been shown to increase label efficiency across several medical tasks, they still require a supervised fine-tuning step after pre-training that requires manually labelled data for the model to predict relevant pathologies[13,14]. As a result, these approaches are only able to predict diseases that were explicitly annotated in the dataset, and are unable to predict pathologies that were not explicitly annotated for training. Thus, for the model to predict a certain pathology with reasonable performance, it must be provided with a substantial number of expert-labelled training examples for that pathology during training. This process of obtaining high-quality annotations of certain pathologies is often costly and time consuming, often resulting in large-scale inefficiencies in clinical artificial intelligence workflows.

In this Article, to address these limitations, we applied a machine-learning paradigm where a model can classify samples during test time that were not explicitly annotated during training[15,16]. We present a zero-shot method using a fully self-supervised-learning procedure that does not require explicit manual or annotated labels for chest X-ray image interpretation to create a model with high performance for the multi-label classification of chest X-ray images. The method, which we call CheXzero, uses contrastive learning, a type of self-supervised learning, with image–text pairs to learn a representation that enables zero-shot multi-label classification. The method can also be considered as a form of natural-language supervision or unsupervised learning[15]. In contrast to previous self-supervised approaches, the method does not require fine-tuning using labelled data. Hence, unlike previous self-supervised approaches, the method requires no labels except for testing, and is able to accurately identify pathologies that were not explicitly annotated. To develop the method, we leveraged the fact that radiology images are naturally labelled through corresponding clinical reports and that these reports can offer a natural source of supervision. We show that the performance of the self-supervised method is comparable to the performance of both expert radiologists and fully supervised methods on unseen pathologies in two independent test datasets collected from two different countries. We also show that the self-supervised method outperforms previous label-efficient approaches on chest X-ray pathology classification, suggesting that explicit labels are not required to perform well on medical-image-interpretation tasks when corresponding reports are available for training. Using chest X-rays as a driving example, the self-supervised method exemplifies the potential of deep-learning methods for learning a broad range of medical-image-interpretation tasks from large amounts of unlabelled data, thereby decreasing inefficiencies in medical machine-learning workflows that result from large-scale labelling efforts.

## Results

We leverage zero-shot learning to classify pathologies in chest X-rays without training on explicit labels (Fig. 1). To do so, we took image–text pairs of chest X-rays and radiology reports, and the model learned to predict which chest X-ray corresponds to which radiology report. We trained the model with 377,110 pairs of a chest X-ray image and the corresponding raw radiology report from the MIMIC-CXR dataset[17].

The performance of the self-supervised model is comparable to that of three benchmark radiologists classifying the five CheXpert competition pathologies evaluated on the CheXpert test dataset. On the Matthews correlation coefficient (MCC) metric, there is no statistically significant difference (model − radiologist

[1]Stanford University Department of Computer Science, Stanford, CA, USA. [2]Department of Biomedical Informatics, Harvard University, Boston, MA, USA. [3]AIMI Center, Stanford University, Palo Alto, CA, USA. [4]These authors contributed equally: Ekin Tiu, Ellie Talius, Pujan Patel. ✉e-mail: pranav_rajpurkar@hms.harvard.edu

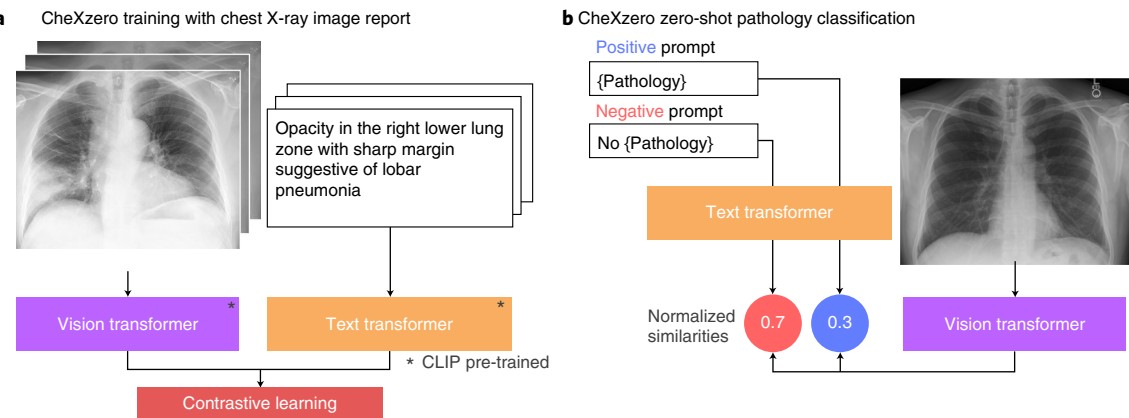

**Fig. 1 | The self-supervised model classifies pathologies without training on any labelled samples. a**, Training pipeline. The model learns features from raw radiology reports, which act as a natural source of supervision. **b**, Prediction of pathologies in a chest X-ray image. For each pathology, we generated a positive and negative prompt (such as 'consolidation' versus 'no consolidation'). By comparing the model output for the positive and negative prompts, the self-supervised method computes a probability score for the pathology, and this can be used to classify its presence in the chest X-ray image.

performance = −0.005; 95% confidence interval (CI) −0.043, 0.034) between the performance of the model (0.523; 95% CI 0.486, 0.561) and that of the radiologists (0.530; 95% CI 0.499, 0.558) averaged over the pathologies. On individual pathologies, the model's MCC performance is higher, but not statistically significantly, compared with radiologists on consolidation (0.018; 95% CI −0.090, 0.123), cardiomegaly (0.058; 95% CI −0.016, 0.133) and oedema (0.015; 95% CI −0.070, 0.099). The model's MCC performance is lower, but not statistically significantly, compared with radiologists on atelectasis (−0.078; 95% CI −0.154, 0.000) and pleural effusion (−0.040; 95% CI −0.096, 0.013). On the F1 metric, there is similarly no statistically significant difference (model − radiologist performance = −0.009; 95% CI −0.038, 0.018) between the mean F1 performance of the model (0.606; 95% CI 0.571, 0.638) and that of the radiologists (0.619; 95% CI 0.585, 0.642) averaged over the pathologies. On individual pathologies, we find that the model F1 performance is significantly higher than that of radiologists on cardiomegaly (model − radiologist performance = 0.065; 95% CI 0.013, 0.115). We find that the model's F1 performance is significantly lower than that of radiologists on atelectasis (model − radiologist performance = −0.045; 95% CI −0.090, −0.001). There are no statistically significant differences in F1 for consolidation (model − radiologist performance = −0.050; 95% CI −0.146, 0.036), oedema (model − radiologist performance = 0.018; 95% CI −0.053, 0.086) and pleural effusion (model − radiologist performance = −0.034; 95% CI −0.078, 0.008). Figure 2 shows the receiver operating characteristic (ROC) curve performance of the model and the radiologist operating points. Table 1 lists the mean performance of the radiologists and the model, and their associated difference with 95% CI.

The results show that the self-supervised model outperforms three previous label-efficient methods (MoCo-CXR, MedAug and ConVIRT) on the CheXpert dataset, using no explicit labels during training. MoCo-CXR and MedAug use self-supervision using only chest X-ray images. Specifically, MoCo-CXR modifies the contrastive learning framework Momentum Contrast (MoCo) for chest X-ray interpretation. MedAug builds on MoCo pre-training by using patient metadata to select positive chest X-ray image pairs for image–image contrastive pre-training. ConVIRT uses chest X-rays along with associated report data to conduct self-supervision. Specifically, ConVIRT jointly trains a ResNet-50 and a Transformer by leveraging randomly sampled text from paired chest X-ray and radiology-report data to learn visual representations. Unlike our approach, these previous works require a small fraction of labelled

data to enable pathology classification. The self-supervised model's mean area under the curve (AUC) of 0.889 outperforms ConVIRT trained on 1% of the labelled data (AUC 0.870), ConVIRT trained on 10% of the labelled data (AUC 0.881), ConVIRT trained on 100% of the labelled data (AUC 0.881), MedAug trained on 1% of the labelled data (AUC 0.810), MoCo-CXR trained on 1% of the labelled data (AUC 0.802), MoCo-CXR trained on 10% of the labelled data (AUC 0.850) and MoCo-CXR trained on 100% of the labelled data (AUC 0.884) (Table 2). Additionally, on the task of classifying plural effusion, the self-supervised model's mean AUC of 0.932 outperforms MoCo-CXR trained on 0.1% of the labelled data (AUC 0.813) and MoCo-CXR trained on 1% of the labelled data (AUC 0.885), MoCo-CXR trained on 10% of the labelled data (AUC 0.920) and MedAug trained on 1% of the labelled data (AUC 0.906) (Table 3)[13,18]. However, the self-supervised model achieves these results without the use of any labels or fine-tuning, thus showing the capability of the model on a zero-shot task.

The flexibility of zero-shot learning enables the self-supervised model to perform auxiliary tasks related to the content found in radiology reports. We applied the self-supervised model to tasks including differential diagnosis using the PadChest dataset, patient sex prediction and chest radiograph projection (anteroposterior versus posteroanterior) prediction[19].

On the task of differential diagnosis on the PadChest dataset, we find that the model achieves an AUC of at least 0.900 on 6 findings and at least 0.700 on 38 findings out of 57 radiographic findings where $n > 50$ in the PadChest test dataset ($n = 39,053$). We obtain high performance on the CheXpert competition pathologies such as pleural effusion, oedema, atelectasis, consolidation and cardiomegaly, with AUCs of 0.958 (95% CI 0.953, 0.963) for pleural effusion, 0.961 (95% CI 0.946, 0.974) for oedema, 0.798 (95% CI 0.780, 0.817) for atelectasis, 0.871 (95% CI 0.851, 0.888) for consolidation and 0.898 (95% CI 0.894, 0.903) for cardiomegaly (Fig. 3). Compared with the performance of the CheXNet model on the PadChest dataset, we observe that the self-supervised model outperformed their approach on three out of the eight selected pathologies, atelectasis, consolidation and oedema, despite using 0% of the labels as compared with 100% in the CheXNet study (Table 4)[20,21].

In addition to the ensembled self-supervised model, we trained a single model using full radiology reports instead of only the impressions section in order to evaluate zero-shot performance on auxiliary tasks such as the prediction of sex. The model trained with full radiology reports achieved an AUC of 0.936 (95% CI 0.910, 0.959) on sex prediction using the prompts 'the patient's sex is male' and

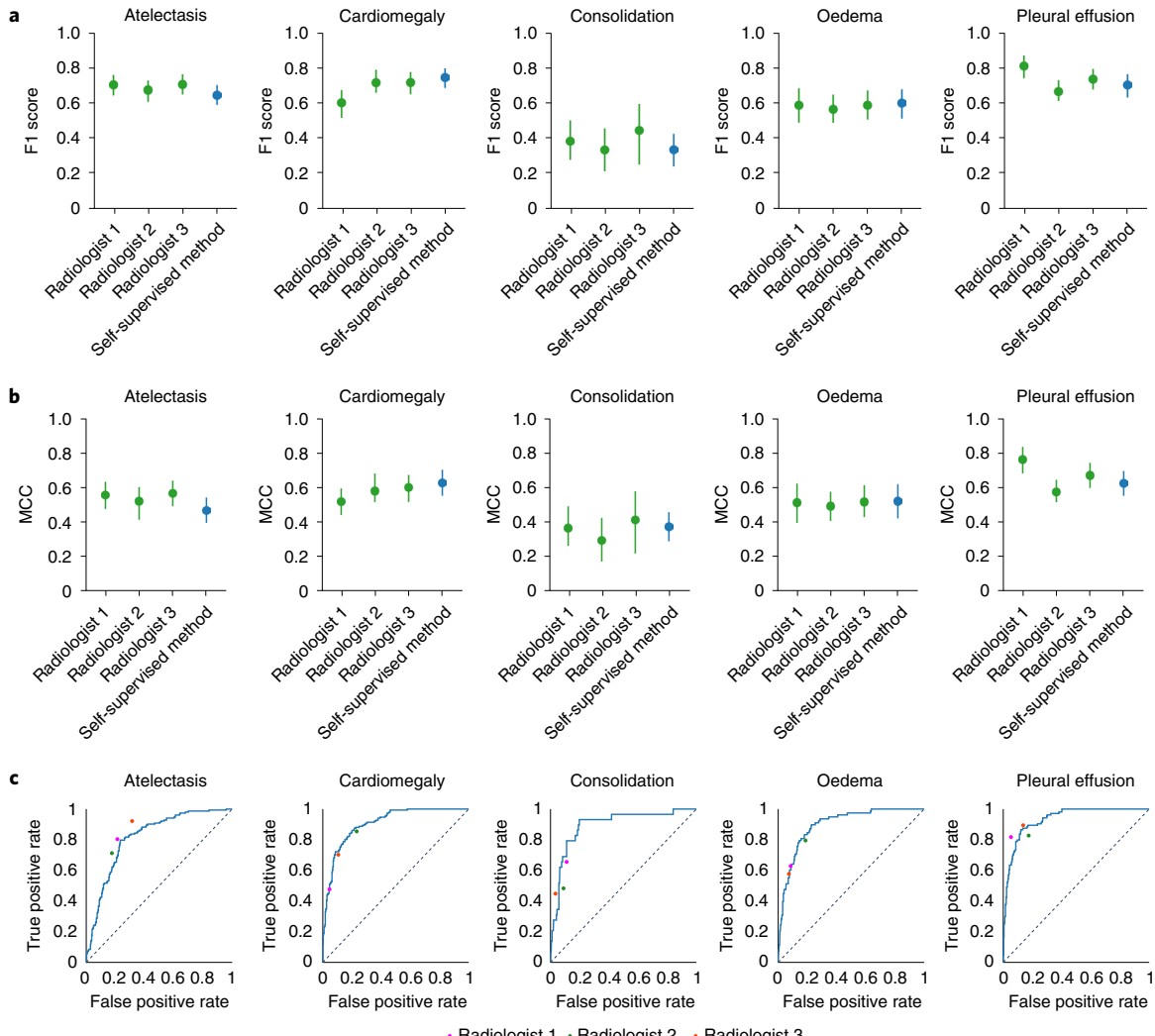

**Fig. 2 | Comparisons of MCC and F1 scores and of ROC curves, for the self-supervised model and board-certified radiologists. a**, F1 scores of the self-supervised model as compared with three board-certified radiologists on the CheXpert test dataset for the five CheXpert competition conditions. The model's F1 score is significantly higher than that of radiologists on pleural effusion, significantly lower on atelectasis and not statistically significantly different on cardiomegaly, consolidation and oedema. **b**, Comparison of the MCC of the self-supervised model against three board-certified radiologists on the CheXpert test dataset. The MCC of the model is not significantly different than that of radiologists on all five pathologies. **a,b**, Green plots indicate the performance of the three board-certified radiologists while blue plots indicate the performance of the self-supervised model. **c**, Comparison of the ROC curve of the self-supervised model to benchmark radiologists against the test-set ground truth. The model outperforms the radiologists when the ROC curve lies above the radiologists' operating points. The dotted lines on the ROC curves represent the baseline performance of a classifier that is no better than random guessing.

'the patient's sex is female'. Additionally, the model achieved an AUC of 0.799 (95% CI 0.7595, 0.835) on the task of predicting whether a chest X-ray is anteroposterior or posteroanterior. To make these predictions on an auxiliary task, the model requires only the development of prompts to use for the task; no training or labels are needed.

## Discussion

The purpose of this work was to develop and demonstrate performance of a zero-shot classification method for medical imaging without training on any explicit manual or annotated labels. The results show that, with no explicit labels, the zero-shot method is comparable to the performance of both expert radiologists and fully supervised methods on pathologies that were not explicitly labelled during training. Specifically, the self-supervised method achieved an AUC −0.042 points below that of the highest-performing fully

supervised model on the CheXpert competition. We also show that the performance of the self-supervised model is comparable to that of radiologists, as there is no statistically significant difference between the performance of the model and the performance of the radiologists on the average MCC and F1 over the five CheXpert competition pathologies. We also show that the self-supervised model outperforms previous label-efficient approaches on chest X-ray pathology classification, suggesting that explicit labels are not required to perform well on medical-image-interpretation tasks when corresponding reports are available for training. We achieved these results using a deep-learning model that learns chest X-ray image features using corresponding clinically available radiology reports as a natural signal. In addition, we show that ensembling over the top-ten highest-performing model checkpoints on the validation dataset can improve the performance of the model (Table 5). We externally validated the self-supervised model, trained on the

**Table 1 | Performance of the self-supervised model, CheXzero, on the five CheXpert competition pathologies in the CheXpert dataset, compared with the performance of three board-certified radiologists**

| | Average | Atelectasis | Cardiomegaly | Consolidation | Oedema | Pleural effusion |
|---|---|---|---|---|---|---|
| **AUC** | | | | | | |
| CheXzero | 0.889 | 0.816 (0.777, 0.852) | 0.906 (0.876, 0.930) | 0.892 (0.823, 0.947) | 0.897 (0.864, 0.928) | 0.932 (0.906, 0.955) |
| **MCC** | | | | | | |
| Radiologists (mean) | 0.530 (0.499, 0.558) | 0.548 (0.496, 0.606) | 0.566 (0.511, 0.620) | 0.359 (0.262, 0.444) | 0.507 (0.431, 0.57) | 0.548 (0.496, 0.606) |
| CheXzero | 0.523 (0.486, 0.561) | 0.468 (0.396, 0.541) | 0.625 (0.553, 0.7) | 0.374 (0.29, 0.458) | 0.520 (0.424, 0.616) | 0.628 (0.558, 0.696) |
| Difference (CheXzero − radiologist) | −0.005 (−0.043, 0.034) | −0.078 (−0.154, 0.000) | 0.058 (−0.016, 0.133) | 0.018 (−0.090, 0.123) | 0.015 (−0.070, 0.099) | −0.04 (−0.096, 0.013) |
| **F1** | | | | | | |
| Radiologists (mean) | 0.619 (0.585, 0.642) | 0.692 (0.646, 0.731) | 0.678 (0.634, 0.718) | 0.385 (0.28, 0.485 | 0.583 (0.511, 0.645) | 0.737 (0.689, 0.783) |
| CheXzero | 0.606 (0.571, 0.638) | 0.646 (0.593, 0.700) | 0.743 (0.685, 0.793) | 0.333 (0.239, 0.424) | 0.602 (0.517, 0.678) | 0.704 (0.634, 0.764) |
| Difference (CheXzero − radiologist) | −0.009 (−0.038, 0.018) | −0.045 (−0.090, −0.001) | 0.065 (0.013, 0.115) | −0.05 (−0.146, 0.036) | 0.018 (-0.053, 0.086) | −0.034 (−0.078, 0.008) |

There is no statistically significant difference between the mean performance of the model and that of the radiologists averaged over the pathologies for MCC and F1. Numbers within parentheses indicate 95% CI.

**Table 2 | Comparison of the self-supervised method, CheXzero, with supervised and self-supervised baseline models on the CheXpert test dataset**

| | Model | Mean AUC |
|---|---|---|
| **Supervised** | DAM | 0.931 |
| | DenseNet-121 | 0.902 |
| **Self-supervised** | GLoRIA[a] | 0.534 |
| | ConVIRT- ResNet-50—1% | 0.870 |
| | ConVIRT- ResNet-50—10% | 0.881 |
| | ConVIRT-ResNet-50—100% | 0.881 |
| | ConVIRT-ViT—1%[b] | 0.725 |
| | ConVIRT-ViT—10%[b] | 0.809 |
| | ConVIRT-ViT—100%[b] | 0.856 |
| | MedAug—1% | 0.810 |
| | MoCo-CXR—1% | 0.802 |
| | MoCo-CXR—10% | 0.850 |
| | MoCo-CXR—100% | 0.884 |
| | **CheXzero—0%** | **0.889** |

Percentages refer to percentage of labels used in the training data. The self-supervised method is only −0.042 points below the highest-performing fully supervised model on the CheXpert competition, Deep AUC Maximization (DAM)[31], and outperforms the self-supervised baselines ConVIRT, MedAug[46] and MoCo-CXR[18]. The mean is over the five selected clinically relevant pathologies in the CheXpert dataset. [a]GLoRIA results were obtained by loading the pre-trained GLoRIA model and performing zero-shot evaluation on the full multi-label CheXpert test dataset. [b]ConVIRT-ViT results were obtained by replacing the ResNet-50 architecture with a Vision Transformer before applying ConVIRT.

**Table 3 | Comparison of the self-supervised ensemble method, CheXzero, with self-supervised baseline models on the CheXpert dataset for the pathology pleural effusion**

| Model | Label fraction | Mean AUC |
|---|---|---|
| MoCo-CXR | 0.1% | 0.813 (0.779, 0.842) |
| MoCo-CXR | 1% | 0.885 (0.856, 0.909) |
| MoCo-CXR | 10% | 0.920 (0.896, 0.941) |
| MoCo-CXR | 100% | 0.953 (0.935, 0.969) |
| MedAug | 1% | 0.906 (0.891, 0.921) |
| CheXzero | 0% | 0.932 (0.906, 0.955) |

The percentages refer to the percentage of labels used in the training data. The self-supervised ensemble model outperforms all self-supervised baseline models that use 10% or less of the data. Numbers within parentheses indicate 95% CI.

MIMIC-CXR dataset, on two independent datasets, the CheXpert test dataset and the human-annotated subset of the PadChest dataset.

Previous efforts for learning with small amounts of labelled data have shown meaningful improvements in performance using fewer labels, but still require the availability of some annotations that may not be trivial to obtain. For instance, recent work has achieved a mean AUC of 0.870 on the CheXpert test dataset using only 1% of the labelled data[14]. However, labelling 1% of a large dataset can still be expensive. For example, 1% of the labelled data in the ChestX-ray14, PadChest and CheXpert datasets amounts to 1,000 labels, 1,609 labels and 2,243 labels, respectively[8,19]. Additionally, these methods can only predict pathologies that were labelled during training, thereby restricting their applicability to other chest pathologies or classification tasks. Therefore, previous label-efficient learning methods may not be as potent in settings where access to a diverse set of high-quality annotations is limited. In contrast, the self-supervised method that we report in this work achieves a mean AUC of 0.889 on the CheXpert test dataset without requiring any explicit annotations (Tables 1 and 2). Recent work has leveraged radiology reports for zero-shot chest X-ray classification; however, it is applicable only to chest X-ray images with only one pathology, limiting the practicality of the method since multiple pathologies are often present in real-world settings[22]. Additionally, recent work has shown that a zero-shot learning approach can predict unseen chest X-ray pathologies, but the method still requires explicit labels

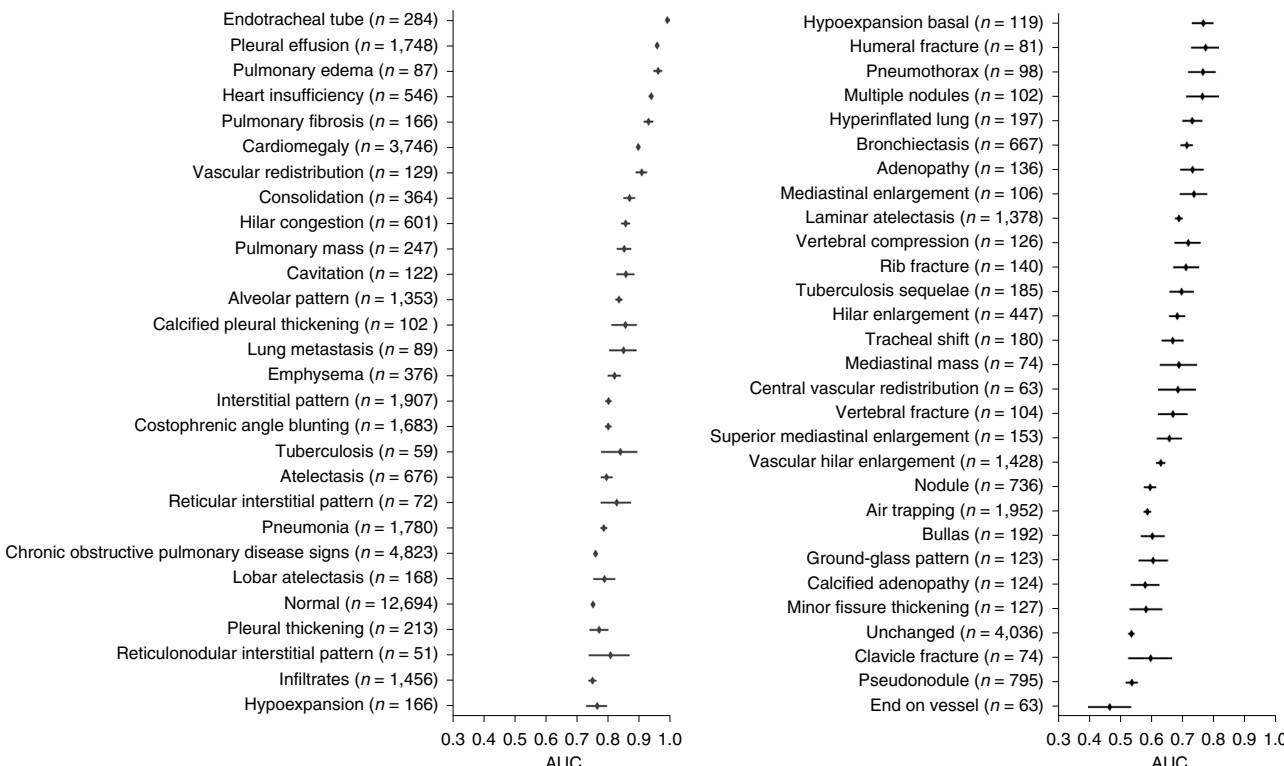

**Fig. 3 | Performance on unseen radiographic findings in the PadChest dataset.** Mean AUC and 95% CI are shown for each radiographic finding ($n > 50$) labelled as high importance by an expert radiologist. We externally validated the model's ability to generalize to different data distributions by evaluating model performance on the human-annotated subset of the PadChest dataset ($n = 39,053$ chest X-rays). No labelled samples were seen during training for any of the radiographic findings in this dataset. The self-supervised method achieves an AUC of at least 0.900 on 6 findings and at least 0.700 on 38 findings out of 57 radiographic findings where $n > 50$ in the PadChest test dataset ($n = 39,053$).

**Table 4 | Comparison of AUC against application of CheXNet[47] with the PadChest dataset**

|  | Atelectasis | Cardiomegaly | Consolidation | Oedema | Lesion | Pneumonia | Pneumothorax | No finding |
|---|---|---|---|---|---|---|---|---|
| CheXNet—100% | 0.794 | 0.908 | 0.840 | 0.939 | 0.707 | 0.806 | 0.873 | 0.871 |
| CheXzero—0% | 0.798 | 0.898 | 0.871 | 0.961 | 0.651 | 0.789 | 0.766 | 0.755 |

The pathologies selected are the pathologies reported in CheXNet[47]. The self-supervised method that is trained on a dataset from a country different from PadChest outperforms CheXNet, a fully supervised model trained on PadChest, on three out of eight pathologies reported in CheXNet: atelectasis, consolidation and oedema. Numbers within parentheses indicate 95% CI.

during training[23]. Our model does not require labels for any pathology since we do not have to distinguish between 'seen' and 'unseen' classes during training.

To increase the number of labelled datasets and to reduce the effort required for manual annotations by domain experts, recent works have designed automatic labellers that can extract explicit labels from unstructured text reports. However, the development time of automatic labelling systems such as the NIH labeller and CheXpert are high, each requiring either extensive domain knowledge or technical expertise to implement[7,24]. This burden is not limited to chest X-rays; previous works have developed labelling methods for several forms of unstructured clinical text such as cancer-pathology reports and electronic health records[25–27]. In contrast, our method is able to classify pathologies without requiring the domain-specific development of an automatic labeller. The self-supervised method has the potential to alleviate the labelling bottleneck in the machine-learning pipeline for a range of medical-imaging tasks by leveraging easily accessible unstructured text data without domain-specific pre-processing efforts[17]. As a

result, the self-supervised method opens promising avenues for approaches and applications in the medical-imaging domain, where narrative reports that describe imaging findings are common.

One notable finding is the ability of the self-supervised method to predict differential diagnoses and radiographic findings with high accuracy on a dataset that was collected in a country different from that of the training dataset[19]. This ability to generalize to datasets from vastly different distributions has been one of the primary challenges for the deployment of medical artificial intelligence[28,29]. Despite the challenges of generalization described in previous works, the self-supervised method achieves an AUC of at least 0.900 on 6 radiographic findings and at least 0.700 on 38 findings out of 57 radiographic findings where $n > 50$ in the PadChest test dataset ($n = 39,053$) (Fig. 3). We speculate that the self-supervised model can generalize better because of its ability to leverage unstructured text data, which contains more diverse radiographic information that could be applicable to other datasets. Additionally, we note that we might expect improved performance if we used alternative labels instead of the raw clinical findings in PadChest. Ultimately, the

**Table 5 | Impact of ensembling on performance**

|  | Mean AUC | Mean F1 | Mean MCC |
|---|---|---|---|
| Radiologists (mean) | N/A | 0.619 | 0.530 |
| Best single model | 0.878 | 0.563 (0.527, 0.598) | 0.473 (0.434, 0.510) |
| Ensemble model | 0.889 | 0.606 (0.571, 0.638) | 0.523 (0.486, 0.561) |

Comparison between the ensemble over top-ten model checkpoints and the single best model on the CheXpert validation dataset. The results were averaged across the five CheXpert competition pathologies. Numbers within parentheses indicate 95% CI. *The Mean AUC of radiologists is not available (N/A) because the binary radiologist predictions are represented by a single point on the receiver operating curve; therefore an area cannot be computed.

results demonstrate that the self-supervised method can generalize well on a different data distribution without having seen any explicitly labelled pathologies from PadChest during training[30].

Biases may have affected the training of the self-supervised method. For example, if a pathology is never mentioned in the reports, then the method cannot be expected to predict that pathology with high accuracy during zero-shot evaluation. Furthermore, the model's ability to predict a pathology may depend on the terminology used in the training reports. For instance, if several reports describe a condition such as atelectasis, but do not explicitly use the term, then the method may not perform well when queried with the phrase 'has atelectasis'[31]. Thus, the method's ability to predict pathologies is limited to scenarios mentioned in the text reports, and may perform less well when there are a variety of ways to describe the same pathology. To address these potential biases, we provide the model with hundreds of thousands of image–text pair samples ($n = 377,110$) during training, encompassing a wide variety of writing styles and descriptions of pathologies[17]. By validating the method on the CheXpert and PadChest datasets, which were collected at different hospitals from the one used in the training of the model, we show that site-specific biases are not inhibiting the method's ability to predict clinically relevant pathologies with high accuracy.

This work has a few limitations. First, the self-supervised method still requires repeatedly querying performance on a labelled validation set for hyperparameter selection and to determine condition-specific probability thresholds when calculating MCC and F1 statistics. Second, the self-supervised method is currently limited to classifying image data; however, medical datasets often combine different imaging modalities, can incorporate non-imaging data from electronic health records or other sources, or can be a time series. For instance, magnetic resonance imaging and computed tomography produce three-dimensional data that have been used to train other machine-learning pipelines[32–34]. On the same note, it would be of interest to apply the method to other tasks in which medical data are paired with some form of unstructured text. For instance, the self-supervised method could leverage the availability of pathology reports that describe diagnoses such as cancer present in histopathology scans[26,35,36]. Lastly, future work should develop approaches to scale this method to larger image sizes to better classify smaller pathologies[37–45].

In summary, we have designed a self-supervised method using contrastive learning that detects the presence of multiple pathologies in chest X-ray images. The self-supervised method builds on the use of image–text pairings of chest X-rays and radiology reports in ConVIRT, as well as on the multi-class zero-shot classification of natural images in Contrastive Language-Image Pre-training (CLIP)[15] to enable the application of zero-shot approaches to medical-image interpretation. The self-supervised method matches radiologist-level performance on a chest X-ray classification task for multiple pathologies that the model was not explicitly trained to classify (Fig. 2 and Table 1). The results highlight the potential of deep-learning models to leverage large amounts of unlabelled data for a broad range of medical-image-interpretation tasks, and thereby may reduce the reliance on labelled datasets and decrease clinical-workflow inefficiencies resulting from large-scale labelling efforts.

## Methods

**Datasets.** *Training.* The self-supervised method was trained on the MIMIC-CXR dataset, a publicly available dataset of chest radiographs with radiology text reports. The MIMIC-CXR dataset contains 377,110 images corresponding to 227,835 radiographic studies[17]. For instances where a radiographic study contains more than one chest X-ray image, the chest X-ray that is in anteroposterior/posteroanterior view was chosen to be included as part of training. Each radiographic study comes with a corresponding free-text radiology report, a summarization written by radiologists regarding their findings. Each full radiology report consists of multiple sections: examination, indication, impression, findings, technique and comparison. CheXpert is a public dataset for chest radiograph interpretation, consisting of 224,316 chest X-rays of 65,240 patients from Stanford Hospital[8]. The dataset is labelled for the presence of 14 different conditions: atelectasis, cardiomegaly, consolidation, oedema, enlarged cardiomediastinum, fracture, lung lesion, lung opacity, no finding, pleural effusion, pleural other, pneumonia, pneumothorax and support devices. These labels are obtained from the agreement of five board-certified radiologists. Additionally, the dataset consists of free-text radiology reports that are associated with each chest X-ray image. The CheXpert validation dataset is utilized for tuning-condition-specific probability thresholds to obtain predictions from the self-supervised model's probabilities for the five CheXpert competition conditions of a given chest X-ray image We conduct this analysis by running inference with the self-supervised model to obtain probability values of each condition being present for all chest X-ray images. Condition-specific probability thresholds are then determined by choosing the probability values that result in the best MCC for each condition on the CheXpert validation dataset. The CheXpert validation dataset has no overlap with the CheXpert test dataset used for evaluation.

*Evaluation.* The self-supervised method was evaluated on two external datasets: the CheXpert test dataset and PadChest. The CheXpert test dataset is a collection of chest X-rays that are commonly used to evaluate the performance of models on chest X-ray interpretation tasks[14,31]. We evaluate the model on the entire CheXpert test dataset, consisting of 500 chest X-ray images labelled for the presence of 14 different conditions[8]. The CheXpert test dataset is utilized to calculate both the self-supervised model's area under the receiver operating characteristic (AUROC) and MCC metrics for each of the five CheXpert competition conditions. Additionally, the test set contains predictions from three board-certified radiologists on full-resolution images with which we compare the performance of the model.

The PadChest dataset is a public dataset that contains 160,868 chest X-ray images labelled with 174 different radiographic findings, 19 differential diagnoses[19]. Twenty-seven per cent of the labels come from board-certified radiologists, and the rest were obtained by using a recurrent neural network with attention trained on the radiology reports. For evaluation purposes, only 39,053 examples from the dataset were utilized, each of which was annotated by board-certified radiologists. These examples were then used to calculate the self-supervised model's AUROC for each of the different conditions described above.

*Pre-processing.* Each of the 377,110 chest X-rays in the MIMIC-CXR dataset were re-sized to $224 \times 224$ and zero padded before training. Each image was then normalized using a sample mean and standard deviation of the training dataset.

Text from radiology reports were tokenized using the byte pair encoding procedure with a vocabulary size of 49,408. For text that exceeds the maximum token sequence length of the given architecture, we truncated the text embedding to the first 'context length tokens – 2'. The remaining two tokens were saved for the [SOS] and [EOS] tokens at the beginning and end of the text embedding, respectively.

**Architecture.** The uninitialized architectures consist of a Vision Transformer, ViT-B/32, for the image encoder, and a Transformer for the text encoder. We use a pre-trained Vision Transformer that accepts images of resolution $224 \times 224$. The text encoder Transformer has a base size of 63 million parameters, 12 layers and a width of 512 with 8 attention heads. The Transformer operates on lower-byte pair encoding representation of text and uses text embeddings with a maximum token length of 77. We use the same initialization scheme used in CLIP[15].

**Implementation of the method.** *Model pre-training.* The self-supervised model consists of an image and text encoder that we jointly train on the MIMIC-CXR

training dataset[17]. We utilize the impressions section of each text report, since it contains a concise summary of the entire report. We contrast this with a previous self-supervised method, ConVIRT, which selects a random sentence from the full-length radiology report for each image[14]. Although their proposed method could extract some signal, a random text input selection allows for unnecessary stochasticity that could lead to inconsistencies in training. To address this, we consistently select the text from the impressions section.

*Training.* We initialized the self-supervised model using the ViT-B/32and Transformer architectures with pre-trained weights from OpenAI's CLIP model[15]. When training on the impressions section, we keep the maximum context length of 77 tokens as given in the CLIP architecture. We demonstrated that we can leverage the pre-trained weights from the CLIP architecture learned from natural images to train a zero-shot model with a domain-specific medical task.

To prepare the data for training, all images from the MIMIC-CXR dataset are stored in a single HDF5 file. We performed a hyperparameter sweep over the batch size and the learning rate using the CheXpert validation dataset. We compute the validation mean AUC over the five CheXpert competition pathologies after every 1,000 batches are trained, and save the model checkpoint if the model outperforms the last best model during training. The validation mean AUCs of these checkpoints are used to select models for ensembling. The best model uses stochastic gradient descent for optimization with a learning rate of 0.0001 and momentum of 0.9. The best model has a batch size of 64 and is trained for four epochs. We train the model by maximizing the cosine similarity between image and text embeddings of all valid image–report pairs in the batch while minimizing the cosine similarity between the embeddings of incorrect pairings in the batch. The method's training procedure closely follows the implementation of CLIP[15].

*Softmax evaluation technique for multi-label classification.* To evaluate the zero-shot performance of the model on the multi-label classification task, we used a positive–negative softmax evaluation procedure on each of the diseases. In contrast to CLIP, the proposed procedure allows us to normalize with respect to the negated version of the same disease classification instead of naively normalizing across the diseases to obtain probabilities from the logits[15]. The latter approach is less reasonable in this context since a single image may have multiple associated labels.

We define the procedure as follows. First, we compute logits with positive prompts (such as atelectasis) and negative prompts (that is, no atelectasis). Then, we compute the softmax between the positive and negative logits. Lastly, we keep the softmax probabilities of the positive logits as the probability that the disease is present in the chest X-ray.

*Ensembling.* We ensemble the top-ten model checkpoints sorted by mean AUC over the five CheXpert pathologies on the validation dataset. The probability outputs of the ensemble are computed by taking the average of the probability outputs of each model. The probabilities are averaged after softmax evaluation. These probabilities are then used for model evaluation through AUC and for prediction tasks using condition thresholds generated from the validation dataset.

*Knowledge-distillation procedure.* To allow for the use of the CLIP pre-trained model on full radiology reports to evaluate zero-shot performance on auxiliary tasks such as sex prediction, we use a knowledge-distillation procedure. This procedure is required as the pre-trained text encoder from the CLIP model has a context length of only 77 tokens, which is not long enough for an entire radiology report. We use the pre-trained model to train a model with a context length of 512, long enough to encompass 98% of radiology reports. In this method, the text encoder of the best-performing model trained only on impressions is used as a teacher for the text encoder of a student model. To train the student, we compute the mean squared error between the logits of the two encoders, then backpropagate across the student architecture. Once the student text encoder is trained, we replace the uninitialized image encoder in the student model with the image encoder of the teacher model. Then, the student model is contrastively trained on the MIMIC-CXR chest X-ray and full-text report pairs.

*Prompt-engineering methods.* We run experiments using the labels present in the test set as the prompts and creating the prompts of '<label>' and 'no <label>' as the positive and negative prompts for the softmax evaluation procedure.

**Statistical analysis.** *AUROC.* We collect AUROC results from both the CheXpert test dataset (500 samples) as well as PadChest dataset (39,053 samples) using the self-supervised model's predictions. The AUROC and MCC results of the five clinically relevant pathologies on the CheXpert test dataset are presented in Table 1. Table 2 consists of the mean AUROC of these five pathologies on the CheXpert test dataset along with self-supervised and supervised comparisons. The DAM supervised method is included as a comparison and currently is state-of-the-art on the CheXpert dataset. An additional supervised baseline, DenseNet121, trained on the CheXpert dataset is included as a comparison since DenseNet121 is commonly used in self-supervised approaches. Current top-performing label-efficient approaches, ConVIRT, MedAug and MoCo-CXR, are included as self-supervised comparisons.

*MCC and F1 score.* To obtain the MCC, we first run inference on the CheXpert test set using our softmax evaluation technique to obtain probability values for the 14 different conditions on each of the 500 chest X-ray images. The probabilities are then transformed into positive/negative predictions using the probability thresholds computed by optimizing MCC over the validation dataset. Then, the condition-based MCC scores are calculated using these predictions. We similarly compute the F1 score, but using the same thresholds as used for computing the MCC.

**Confidence intervals.** We use the non-parametric bootstrap to generate confidence intervals: random samples of size $n$ (equal to the size of the original dataset) are repeatedly sampled 1,000 times from the original dataset with replacement. We then estimate the AUROC, F1 and MCC metrics (or their difference for two the methods) using each bootstrap sample. We derive confidence intervals from the relative frequency distribution of the estimates over the re-samples, using the interval between the $100 \times (\alpha/2)$ and $100 \times (1 - \alpha/2)$ percentiles; we pick $\alpha = 0.05$.

**Reporting summary.** Further information on research design is available in the Nature Research Reporting Summary linked to this article.

## Data availability

The main data (CheXpert data) supporting the results of this study are available at https://aimi.stanford.edu/chexpert-chest-x-rays. MIMIC-CXR data are available at https://physionet.org/content/mimic-cxr/2.0.0 for users with credentialed access. PadChest data are available at https://bimcv.cipf.es/bimcv-projects/padchest. Source data are provided with this paper.

## Code availability

The code used to train and evaluate CheXzero is available on GitHub at https://github.com/rajpurkarlab/CheXzero.

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

## Acknowledgements

The authors acknowledge the contributions of the consortium working on the development of the NHLBI BioData Catalyst ecosystem.

## Author contributions

P.R, A.Y.N., E. Tiu, E. Talius. and P.P. conceptualized the study. P.R., E. Tiu, P.P. and E. Talius designed the study. E. Tiu., E. Talius, P.P., P.R. and C.L. performed data analysis and interpretation. E. Tiu, E. Talius, P.P. and P.R. drafted the manuscript. A.Y.N. and C.L. carried out critical revisions of the manuscript, with important intellectual content. A.Y.N. and P.R. supervised the work.

## Competing interests

The authors declare no competing interests.

## Additional information

**Correspondence and requests for materials** should be addressed to Pranav Rajpurkar.

# Reporting Summary

## Statistics

For all statistical analyses, confirm that the following items are present in the figure legend, table legend, main text, or Methods section.

| n/a | Confirmed | |
|---|---|---|
| ☐ | ☒ | The exact sample size (*n*) for each experimental group/condition, given as a discrete number and unit of measurement |
| ☐ | ☒ | A statement on whether measurements were taken from distinct samples or whether the same sample was measured repeatedly |
| ☒ | ☐ | The statistical test(s) used AND whether they are one- or two-sided <br> *Only common tests should be described solely by name; describe more complex techniques in the Methods section.* |
| ☒ | ☐ | A description of all covariates tested |
| ☒ | ☐ | A description of any assumptions or corrections, such as tests of normality and adjustment for multiple comparisons |
| ☒ | ☐ | A full description of the statistical parameters including central tendency (e.g. means) or other basic estimates (e.g. regression coefficient) AND variation (e.g. standard deviation) or associated estimates of uncertainty (e.g. confidence intervals) |
| ☒ | ☐ | For null hypothesis testing, the test statistic (e.g. *F*, *t*, *r*) with confidence intervals, effect sizes, degrees of freedom and *P* value noted <br> *Give P values as exact values whenever suitable.* |
| ☒ | ☐ | For Bayesian analysis, information on the choice of priors and Markov chain Monte Carlo settings |
| ☒ | ☐ | For hierarchical and complex designs, identification of the appropriate level for tests and full reporting of outcomes |
| ☒ | ☐ | Estimates of effect sizes (e.g. Cohen's *d*, Pearson's *r*), indicating how they were calculated |

*Our web collection on statistics for biologists contains articles on many of the points above.*

## Software and code

Policy information about availability of computer code

| Data collection | No software was used to collect the data. |
|---|---|
| Data analysis | The code used to train and evaluate CheXzero is available on GitHub at https://github.com/rajpurkarlab/CheXzero. We developed custom Python code (version 3.6) using the following libraries: Pillow (version 2.2.2, https://pillow.readthedocs.io/en/stable), matplotlib (version 3.3.4, https://matplotlib.org), and numpy (version 1.19.5, https://numpy.org/doc) to visualize and pre-process chest X-ray images. |

For manuscripts utilizing custom algorithms or software that are central to the research but not yet described in published literature, software must be made available to editors and reviewers. We strongly encourage code deposition in a community repository (e.g. GitHub). See the Nature Portfolio guidelines for submitting code & software for further information.

## Data

Policy information about availability of data

All manuscripts must include a data availability statement. This statement should provide the following information, where applicable:

- Accession codes, unique identifiers, or web links for publicly available datasets
- A description of any restrictions on data availability
- For clinical datasets or third party data, please ensure that the statement adheres to our policy

The main data (CheXpert data) supporting the results of this study are available at https://aimi.stanford.edu/chexpert-chest-x-rays. MIMIC-CXR data are available at

## Human research participants

Policy information about studies involving human research participants and Sex and Gender in Research.

| | |
|---|---|
| Reporting on sex and gender | The datasets used in this study have sex distributions publicly reported, with the exception of the CheXpert test dataset, which has been hidden for the purposes of official evaluation on the CheXpert leaderboard. We do not report results stratified by sex; however, the design of the study does explore the ability of the method to predict sex from chest X-rays in the PadChest dataset. We report that, for the final labelled PadChest dataset, 80,923 images correspond to women and 79,923 images to men. |
| Population characteristics | The CheXpert dataset consists of chest radiographic examinations from Stanford Hospital, performed between October 2002 and July 2017 in both inpatient and outpatient centers. Population-level characteristics are unavailable for the CheXpert test dataset, as they are used for official evaluation on the CheXpert leaderboard. <br><br> The MIMIC-CXR and PadChest are datasets available in the public domain. <br><br> MIMIC-CXR is a large dataset involving 65,379 patients imaged at the Beth Israel Deaconess Medical Center Emergency Department during 2011–2016. A total of 377,110 images are available, and are paired with corresponding free-text radiology reports. Each imaging study may contain a frontal and a lateral view. <br><br> The PadChest dataset contains chest X-rays that were interpreted by 18 radiologists at the Hospital Universitario de San Juan, Alicante, Spain, from January 2009 to December 2017. The dataset contains 109,931 image studies and 168,861 images. PadChest also contains 206,222 study reports. The PadChest study reports that the patients' ages range from 0 to 105 years, with a mean of 58.5 years and a median of 62 years. The distribution of the number of images by age is skewed towards older ages, with a long tail for ages under 40. The median birth year of the population was 1953 (birth years ranged from 1904 to 2017), with a standard deviation of 20 years. <br><br> Additional dataset characteristics of the MIMIC-CXR and PadChest datasets are detailed in, respectively, https://www.ncbi.nlm.nih.gov/pmc/articles/PMC6908718 and https://arxiv.org/abs/1901.07441. |
| Recruitment | No participants were recruited for this retrospective study. |
| Ethics oversight | The study used data collected retrospectively. Approval of a study protocol was not needed. |

Note that full information on the approval of the study protocol must also be provided in the manuscript.

## Field-specific reporting

Please select the one below that is the best fit for your research. If you are not sure, read the appropriate sections before making your selection.

☒ Life sciences    ☐ Behavioural & social sciences    ☐ Ecological, evolutionary & environmental sciences

For a reference copy of the document with all sections, see nature.com/documents/nr-reporting-summary-flat.pdf

## Life sciences study design

All studies must disclose on these points even when the disclosure is negative.

| | |
|---|---|
| Sample size | We used 377,110 chest X-ray images corresponding to 227,835 radiographic studies for model training. For the test set, we used the full CheXpert test dataset, consisting of 500 chest X-ray images labelled for the presence of 14 different conditions. Furthermore, we evaluated the model on 39,053 examples from the PadChest dataset, each of which were annotated by board-certified radiologists. We report results on diagnoses where n > 50. |
| Data exclusions | No data were excluded. |
| Replication | The code and data used to train and evaluate CheXzero, which are publicly available (as detailed in the Data-availability and Code-availability statements), can be used to replicate the findings. |
| Randomization | We didn't require randomization, as no human-subject evaluation was performed. |
| Blinding | Blinding wasn't relevant to the study, as no human-subject evaluation was performed. |

## Reporting for specific materials, systems and methods

We require information from authors about some types of materials, experimental systems and methods used in many studies. Here, indicate whether each material, system or method listed is relevant to your study. If you are not sure if a list item applies to your research, read the appropriate section before selecting a response.

## Materials & experimental systems

| n/a | Involved in the study |
|-----|----------------------|
| ☒ ☐ | Antibodies |
| ☒ ☐ | Eukaryotic cell lines |
| ☒ ☐ | Palaeontology and archaeology |
| ☒ ☐ | Animals and other organisms |
| ☒ ☐ | Clinical data |
| ☒ ☐ | Dual use research of concern |

## Methods

| n/a | Involved in the study |
|-----|----------------------|
| ☒ ☐ | ChIP-seq |
| ☒ ☐ | Flow cytometry |
| ☒ ☐ | MRI-based neuroimaging |

