## [Peer Review File · Nature Biomedical Engineering]

Expert-level detection of pathologies from unannotated chest X-ray images via self-supervised learning

Corresponding author: Pranav Rajpurkar

Editorial note

This document includes relevant written communications between the manuscript's corresponding author and the editor and reviewers of the manuscript during peer review. It includes decision letters relaying any editorial points and peer-review reports, and the authors' replies to these (under 'Rebuttal' headings). The editorial decisions are signed by the manuscript's handling editor, yet the editorial team and ultimately the journal's Chief Editor share responsibility for all decisions.

Any relevant documents attached to the decision letters are referred to as **Appendix #**, and can be found appended to this document. Any information deemed confidential has been redacted or removed. Earlier versions of the manuscript are not published, yet the originally submitted version may be available as a preprint. Because of editorial edits and changes during peer review, the published title of the paper and the title mentioned in below correspondence may differ.

Correspondence

Mon 20 Dec 2021

Decision on Article nBME-21-2667

Dear Dr Rajpurkar,

Thank you again for submitting to *Nature Biomedical Engineering* your manuscript, "Pathology Classification on Chest X-rays without Expert Annotation via Self-Supervised Learning". The manuscript has been seen by three experts, whose reports you will find at the end of this message.

You will see that the reviewers appreciate the work, and that they raise a number of methodological queries and provide useful suggestions for improvement. We hope that with further work you can make the methodology clearer and address the reviewers' criticisms. In particular, we would expect that a revised version of the manuscript provides:

* For the pathologies assessed in Fig. 2, performance comparison with a self-supervised learning model relying on explicit annotation labels.

*Thorough description of the model's architecture and implementation.

* Discussion of the processes that you have implemented to avoid any potential biases in the training of the model, such as information related to ground-truth pathology labels that may be available in the free-text clinical reports.

For your information: at least one of the reviewers tested the code, noting that it runs smoothly on the validation set of CheXpert; yet they could not verify the performance numbers provided in the manuscript, because the ground-truth labels of the test set are not publicly available.

When you are ready to resubmit your manuscript, please upload the revised files, a point-by-point rebuttal tothe comments from all reviewers, the reporting summary, and a cover letter that explains the main improvements included in the revision and responds to any points highlighted in this decision.

Please follow the following recommendations:

- * Clearly highlight any amendments to the text and figures to help the reviewers and editors find and understand the changes (yet keep in mind that excessive marking can hinder readability).
- * If you and your co-authors disagree with a criticism, provide the arguments to the reviewer (optionally, indicate the relevant points in the cover letter).
- * If a criticism or suggestion is not addressed, please indicate so in the rebuttal to the reviewer comments and explain the reason(s).
- * Consider including responses to any criticisms raised by more than one reviewer at the beginning of the rebuttal, in a section addressed to all reviewers.
- * The rebuttal should include the reviewer comments in point-by-point format (please note that we provide all reviewers with the reports as they appear at the end of this message).
- * Provide the rebuttal to the reviewer comments and the cover letter as separate files.

We hope that you will be able to resubmit the manuscript within 15 weeks from the receipt of this message. If this is the case, you will be protected against potential scooping. Otherwise, we will be happy to consider a revised manuscript as long as the significance of the work is not compromised by work published elsewhere or accepted for publication at *Nature Biomedical Engineering*.

We hope that you will find the referee reports helpful when revising the work. Please do not hesitate to contact me should you have any questions.

Best wishes,

Pep

Pep Pàmies
Chief Editor, Nature Biomedical Engineering

Reviewer #1 (Report for the authors (Required)):

The paper presents an approach to CXR classification based on self-supervised zero-shot learning. The idea of using radiology reports to generate prompts for CXR images is new. On the other, self-supervised learning for network pre-training for CXR classification has already been used several times. The approach to generate prompts is based on zero-shot learning, where the training is utilized through attributes that are generated for the image. Having said that, authors should highlight what the methodological (or other) novel contribution is. I acknowledge that the presented work has strong translational contribution.

Some remarks about methodology/experimental section.

Chexpert dataset was used to determine condition-specific thresholds. However, then again, Chexpert was used for evaluation. Is this the correct approach? What about possible positive bias? How were the 500 images from chexpert selected? Are these subset of images used for training?
The second dataset used for evaluation was PadChest. Again, the authors selected only 2978 images. Why? Why not use all 27% of images that have labels available.
It would be interesting to compare with some of the recently proposed state-of-the art (self-supervised) approaches, not only comparison with the baseline model.

Hints:

Table 3 is low quality, hard to read.

Is it necessary to define AUROC and MCC? These are notoriously known.

Some brief overview of similar works can be beneficial to the reader

Reviewer #2 (Report for the authors (Required)):

The proposed approach can match radiologists' performance on multi-label pathology classification of chest X-rays and can generalize to pathologies that were not explicitly annotated for training. Additionally, the approach outperforms a fully-supervised model on three out of eight pathologies on an external validation set obtained from a hospital in a different country.

Overall, this is a paper demonstrating the potential of a previously published machine learning method (CLIP) [1] in pathology classification of chest x-rays. My concerns mainly come from its limited technical novelty (i.e., minor modifications on top of CLIP) and its design of the prompt engineering method. I consider the degree of advance of this work to be translational.

[1] Radford et al. Learning Transferable Visual Models From Natural Language Supervision. Proceedings of the 38th International Conference on Machine Learning, PMLR 139:8748-8763.

I have the following concerns and comments.

1. One of my major concerns is that the proposed method seemed to be largely built upon CLIP (Contrastive Language-Image Pre-training) [1] (I also suggest to replace reference 15 with a more formal citation as CLIP has been published at ICML 2021). As far as I am concerned, the only obvious modification is that the authors replaced ResNet used in CLIP with ViT-B/32. Also, the authors did not give clarifications about the differences between the proposed approach and CLIP, which makes me feel the technical contribution of the proposed method is quite limited.

2. (Mandatory) Another major concern is that the proposed prompt engineering procedure in the Methods section seems to have serious flaws. The main steps in this procedure include 1) "A board-certified radiologist developed a list of 73 alternative prompts." 2) "We then used the standard template of "___" and "not ___" with each of the alternative labels and had the model predict the probability of the image corresponding to each label. Then, we were able to determine the best performing label for each pathology by looking at the AUC for each." Here in the second step, computing AUC requires groundtruth pathology labels. It is not clear to me whether this prompt engineering procedure uses the CheXpert test dataset or a validation subset of the CheXpert training set. If the test set had been used, the authors would have used the groundtruth pathology labels of the test set to choose the best performing prompts, which should not be allowed. If a validation subset of the training set had been used, the authors would have used the groundtruth labels of the validation subset, which means the proposed method is "supervised" and the claims about zero-shot learning do not hold any more. Therefore, the proposed prompt engineering procedure is quite problematic either way.

3. (Mandatory) Furthermore, the proposed method requires much effort from a board-certified radiologist to develop alternative prompts for every pathology. Although this effort does not directly assign pathology labels to individual x-ray images, it is a manual process and requires the domain knowledge of a human expert. Therefore, calling the proposed method "self-supervised" is not very rigorous. In addition, radiology reports were also originally produced by human experts although for a different purpose. I believe it would be less misleading if the authors can use terms like text-supervised or report-supervised learning to distinguish text-level supervision from image-level supervision.

4. (Mandatory) Why did you need to "acquire free-text radiology reports corresponding to each of the 500 chest x-ray images to enable zero-shot evaluation"? These are the radiology reports of the CheXpert test dataset, and they have information related to the groundtruth pathology labels. Exploiting such information during performance evaluation on the test set should not be allowed.

5. (Mandatory) All figures in the paper have low resolution, making it difficult to see more details.

6. (Mandatory) Legends are missing in Fig. 2c. What do those points with 3 different colors stand for? Do they represent 3 radiologists? Then, why do their performance have so large gaps?
7. Comma is missing in the caption of Fig. 3 (2978->2,978).
8. (Mandatory) In experimental results, what does DAN in the caption of Table 2 stand for? A reference should be added here. In Table 2, the authors used DAM. Do DAN and DAM have the same meaning or not?

Reviewer #3 (Report for the authors (Required)):

The aim of this paper is to develop a SSL based method without explicit annotation labels to outperform a fully supervised learning with explicit annotation labels, which can match radiologists's performance on multi-label pathology classification with CXR. This is quite interesting topics. However, there are several concerns on the reproducibility of this study as follows.

1. The size of CXR (320x320) is too small to find lung nodule, which should be read in CXR to catch There is a lack of sample dataset. The size of CXR should be 512 x 512 or 1K x 1K to find the nodule in CXR.
2. For radiologists' reading study, what is the size of CXR?
3. Why do you select 5 pathologies in the CheXpert. In addition, other classes from 14 pathologies in the CheXpert should be declared.
4. The authors insist that their method outperform the supervised learning. However, there was only comparison to semi-supervised learning in Table 2, and 3. In this table, Mean AUC of supervised learning (0.931) is better than this method (0.915).
5. There is a lack of details on this paper to authors considering the differences of semi-supervised methods in table 2 and 3, and other classes' result in table 3.

Wed 02 Feb 2022

Decision on Article nBME-21-2667A

Dear Dr Rajpurkar,

Thank you again for submitting to *Nature Biomedical Engineering* your revised manuscript, "Pathology Classification on Chest X-rays without Expert Annotation via Self-Supervised Learning".

The manuscript has been seen by the three original experts, whose reports you will find at the end of this message. In particular, you will see that Reviewer #2 states that tuning the hyperparameters using a subset of the training data disqualifies the approach as a zero-shot method, and that selecting best-performing prompts from the test set of CheXpert may not be appropriate. These criticisms impinge directly on the main claims of the work, and it is unclear to us whether you would be able to address them. Hence, I return the manuscript to you so that you can decide how to proceed. We would only invite a revision if the claimed performance and methodology advantages can be upheld.

We hope that you will find the referee reports helpful when revising the work.

Best wishes,

Pep

Pep Pàmies
Chief Editor, Nature Biomedical Engineering

* Although we cannot publish your paper, it may be appropriate for another journal in the Nature Portfolio. If you wish to explore the journals and transfer your manuscript please use our manuscript transfer portal. If you transfer to Nature journals or the Communications journals, you will not have to re-supply manuscript metadata and files. This link can only be used once and remains active until used.

All Nature Portfolio journals are editorially independent, and the decision on your manuscript will be taken by their editors. For more information, please see our manuscript transfer FAQ page.

Note that any decision to opt in to *In Review* at the original journal is not sent to the receiving journal on transfer. You can opt in to In Review at receiving journals that support this service by choosing to modify your manuscript on transfer. *In Review* is available for primary research manuscript types only.

Reviewer #1 (Report for the authors (Required)):

The authors answered/explained in the manuscript most of my remarks.

I would like again raise the issue of novelty, especially in the context of some previous works such as:

1. Smit, Akshay, Saahil Jain, Pranav Rajpurkar, Anuj Pareek, Andrew Y. Ng, and Matthew P. Lungren. 436 2020. —CheXbert: Combining Automatic Labelers and Expert Annotations for Accurate Radiology Report 437 Labeling Using BERT.
2. Vu, Yen Nhi Truong, Richard Wang, Niranjana Balachandar, Can Liu, Andrew Y. Ng, and 455 Pranav Rajpurkar. 2021. —MedAug: Contrastive Learning Leveraging Patient Metadata Improves 456 Representations for Chest X-Ray Interpretation.

-The author explained novelty in general but not in the context of the previous published works of some of the co-authors.

Another issue that I am not completely satisfied with is the review of the recent works. The authors add some recent papers, but mostly arxiv published papers. It would be better if the authors included preferably peer-reviewed papers.

Reference 13 and 19 in manuscript is the same one.

Otherwise, this is very solid work and an interesting application of ML.

Reviewer #2 (Report for the authors (Required)):

Thanks for the authors' detailed responses. However, two major concerns still remain and I have an additional concern about the inconsistency of the reported experimental results.

1. The authors' responses as well as lines 269-274 and lines 327-329 in the revised manuscript confirmed that the validation set of CheXpert had been used for tuning hyperparameters, including probability thresholds. I would like to point out that the validation set is actually a part of training data, and hyperparameter tuning needs to access the labels of the validation set. This is contradictory to the most important claim that the proposed method is a zero-shot method, which is not supposed to touch any labels in the training data at all.
2. The authors' responses and the captions of Tables 6 and 8 also confirmed that the majority of experimental results (including the results in Tables 1 and 2) on the CheXpert test set were obtained using best performing alternative prompts evaluated on the CheXpert test set itself. In particular, Table 1 shows the main results of the proposed CheXzero approach. Only Tables 7 and 8 report results obtained using default prompts. Note that using the test set to select best performing prompts is unacceptable even as a future avenue for exploration.
3. The results reported in Tables 6 and 8 are inconsistent. As stated by the authors, the results in Tables 6 and 8 are reported on the CheXpert test set. Thus, the improvements achieved with the best performing alternative prompts over the default prompts should be the same in both tables. However, I noticed that at least half of them are not consistent. For example, over Consolidation and Edema, Table 6 shows that the best performing alternative prompts achieve 0.039 and 0.019 improvements in AUC, but in Table 8, the alternative prompts do not bring any improvements. In contrast, the results on Atelectasis and Cardiomegaly are consistent.

Reviewer #3 (Report for the authors (Required)):

Thank for your kind response. However, the size of CXR could be critical for clinical application of your methods.

Tue 12 Apr 2022

Decision on Article nBME-21-2667B-Z

Dear Dr Rajpurkar,

Thank you for your latest version of the manuscript, "Pathology Classification on Chest X-rays without Expert Annotation via Self-Supervised Learning", which has been seen by Reviewer #2. In their report, which you will find at the end of this message, you will see that the reviewer raises a few additional technical criticisms that I hope you will be able to address. Because for this work our editorial emphasis has been on advantageous performance, we find the reviewer's criticisms to be particularly relevant.

As before, when you are ready to resubmit your manuscript, please upload the revised files, a point-by-point rebuttal to the comments from the reviewer, and the reporting summary.

We look forward to receive a further revised version of the work. Please do not hesitate to contact me should you have any questions.

Best wishes,

Pep

Pep Pàmies
Chief Editor, Nature Biomedical Engineering

Reviewer #2 (Report for the authors (Required)):

My previous concerns have been addressed in this revised version. However, I have the following new concerns.

- 1) The main results of the proposed method reported in this version was obtained using an ensemble of 10 models while the performance of those compared self-supervised methods was obtained from single models. Such comparisons are unfair. I notice the mean AUC of the best single model of the proposed method, as shown in Table 7, is 0.878, which is almost the same as the performance of ConVIRT (0.870) when 1% training data is used.
- 2) The proposed method uses ViT as the backbone while ConVIRT uses ResNet50 as the backbone for image feature extraction (Transformer is used for text reports not for images). This is also unfair as ViT is more powerful than ResNet50.

I would like to see a comparison where both ConVIRT and the proposed method use the same backbone for image feature extraction and their performance is measured on single models. To warrant publication in NBME, the proposed zero-shot method should clearly outperform ConVIRT trained with 1% training data.

Sat 16 Apr 2022

Decision on Article nBME-21-2667C

Dear Dr Rajpurkar,

Thank you for your latest version of the manuscript, "Pathology Classification on Chest X-rays without Expert Annotation via Self-Supervised Learning", which has been seen by Reviewer #2. In their report, which you will find at the end of this message, you will see that the reviewer insists, by providing reasonable arguments, in that the current comparisons with the other self-supervised models may not be considered fully fair. I hope that an agreement can be reached regarding which comparisons are most appropriate; in particular, please let me know whether at least one of the suggested comparisons in the reviewer's point #3 can be implemented.

As before, when you are ready to resubmit your manuscript, please upload the revised files, a point-by-point rebuttal to the comments from the reviewer.

Best wishes,

Pep

Pep Pàmies
Chief Editor, Nature Biomedical Engineering

Reviewer #2 (Report for the authors (Required)):

I have to disagree with the authors' latest responses.

1) Ensembling could be added to many machine learning algorithms for the purpose of performance boosting. If ensembling has to be used, for the sake of fairness, it should be added to all algorithms participating in a comparison.

2) The authors wrote that ConVIRT includes a ResNet50 model pre-trained on the ImageNet dataset; a text model that uses the BERT architecture that is initialized with the ClinicalBERT model pre-trained on the MIMIC clinical notes; and some data augmentation strategies. I notice that the ImageNet dataset has around 1 million training images and the MIMIC-CXR dataset has 217k image-text pairs. However, the authors did not mention that the pretrained ViT model they took from CLIP was trained on 400 million image-text pairs collected from the internet. It is obvious that the authors' model has unfair advantages, including a more powerful backbone and a much larger pretraining dataset. If the proposed method cannot switch its backbone to ResNet50, in a fair comparison, ConVIRT should be revised to replace its backbone with the pretrained ViT model used in the current manuscript.

3) The proposed method should be compared with the following two state-of-the-art papers in a fair manner (the same backbone and the same setting regarding model ensembling), and should demonstrate clearly better performance. Although the first paper (ConVIRT) is still in preprint format, it was posted on arXiv about 18 months ago in October 2020 and has been widely regarded as a state-of-the-art method. The second paper is a published ICCV 2021 paper, but is not compared with in the current manuscript.

Yuhao Zhang, Hang Jiang, Yasuhide Miura, Christopher D Manning, and Curtis P Langlotz. Contrastive learning of medical visual representations from paired images and text. arXiv preprint arXiv:2010.00747, 2020.

Shih-Cheng Huang, Liyue Shen, Matthew P. Lungren, and Serena Yeung. GLoRIA: A Multimodal Global-Local Representation Learning Framework for Label-efficient Medical Image Recognition. ICCV 2021.

Wed 29 Jun 2022

Decision on Article nBME-21-2667D

Dear Dr Rajpurkar,

Thank you for the latest version of your revised manuscript, "Pathology Classification on Chest X-rays without Expert Annotation via Self-Supervised Learning". Having consulted with Reviewer #2 (whose comments you will find at the end of this message), I am pleased to write that we shall be happy to publish the manuscript in *Nature Biomedical Engineering*.

We will be performing detailed checks on your paper and will send you a checklist detailing our editorial and formatting requirements in due course.

Best wishes,

Pep

Pep Pàmies
Chief Editor, Nature Biomedical Engineering

Reviewer #2 (Report for the authors (Required)):

The latest comparisons with GLoRIA and ConVIRT-ViT look quite good. This manuscript is almost ready. But I didn't find any code in this round of submission. Please submit your latest code, including the ConVIRT-ViT code and model. Thanks.

[After re-supply of a complete code]

I have checked the code at this new link, and everything looks reasonable.
I have no further comments on this paper.

Rebuttal 1

Review Response

Dear editorial team and reviewers, Thank you for your very helpful feedback! We have responded to your suggestions with additional analyses and explanations, and have updated our manuscript and supplementary material with relevant edits (with Track Changes on). We include our point by point responses below in blue text.

Reviewer #1 (Report for the authors (Required)):

The paper presents an approach to CXR classification based on self-supervised zero-shot learning. The idea of using radiology reports to generate prompts for CXR images is new. On the other, self-supervised learning for network pre-training for CXR classification has already been used several times. The approach to generate prompts is based on zero-shot learning, where the training is utilized through attributes that are generated for the image. Having said that, authors should highlight what the methodological (or other) novel contribution is. I acknowledge that the presented work has strong translational contribution.

Response:

We have now updated the manuscript to highlight what the methodological contribution is. We detail the main distinction between prior self-supervised pre-training methods and our method below, and have made this distinction clearer in the *Main* section.

Prior works use self-supervised pre-training as a means of learning generalizable features from unlabeled data. In prior works, however, a fine-tuning stage which *requires labeled data* must follow the self-supervised pre-training stage in order for the model to predict relevant chest x-ray pathologies. In contrast, our method does not have a fine-tuning stage in which labeled data is required. This has two major implications.

The first is that our method is a zero-shot approach. Unlike prior methods such as [1-3], our method does not require *any* explicit annotations for training, thereby alleviating domain-specific bottlenecks associated with obtaining high quality pathology labels. The second is that our method is able to predict pathologies that were not explicitly annotated. In contrast, prior works are limited to predicting a predefined set of pathologies selected by the ML practitioner that the model observes in the supervised, fine-tuning stage.

“Here we present a novel zero-shot method using a fully self-supervised learning procedure that does not require explicit manual or annotated labels for chest x-ray image interpretation to create a model with high performance on classification of chest x-ray images. Our method, which we call CheXzero, uses contrastive learning, a type of self-supervised learning, with image-text pairs to learn a representation that enables zero-shot classification. Our method can also be considered as a form of natural language supervision or unsupervised learning¹⁵. In contrast to prior self-supervised approaches, our method does not require fine-tuning using labeled data. Thus, unlike prior self-supervised approaches, our method requires no labels

except for testing, and is able to identify pathologies accurately that were not explicitly annotated.”

Additionally, we note that our zero-shot approach, without any explicitly annotated data, matches or outperforms prior self-supervised pre-training approaches such as MedAug and MoCo-CXR. We conducted **two additional experiments comparing the performance of our model to MedAug and MoCo-CXR**. We show that across all label percentages, we outperform prior self-supervised learning methods on the Mean AUC. We include these additional results in Supplemental Table 2, included below for your convenience.

Table 2 | Comparing the self-supervised method to supervised and self-supervised baselines on the CheXpert data set. Percentages refer to percentage of labels used in the training data. The self-supervised method nearly matches the supervised baseline is only -0.038 points below the highest performing fully supervised model on the CheXpert competition, Deep AUC Maximization (DAM)⁴⁶. DAN and outperforms the self-supervised baselines ConVIRT, MedAug¹⁹, and MoCo-CXR¹⁸. Note the mean is over the 5 selected clinically relevant pathologies in the CheXpert data set.

	Model	Mean AUC
Supervised	DAM	0.931
	DenseNet-121	0.902
Self-Supervised	ConVIRT - 1%	0.870
	ConVIRT - 10%	0.881
	ConVIRT - 100%	0.881
	MedAug- 1%	0.810
	MoCo-CXR- 1%	0.802
	MoCo-CXR- 10%	0.850
	MoCo-CXR- 100%	0.884
	Ours - 0%	0.893

[1] Vu, Yen Nhi Truong, Richard Wang, Niranjana Balachandar, Can Liu, Andrew Y. Ng, and Pranav Rajpurkar. 2021. “MedAug: Contrastive Learning Leveraging Patient Metadata Improves Representations for Chest X-Ray Interpretation.” arXiv Preprint arXiv:2102.10663.

[2] Zhang, Yuhao, Hang Jiang, Yasuhide Miura, Christopher D. Manning, and Curtis P. Langlotz. 2020. “Contrastive Learning of Medical Visual Representations from Paired Images and Text.” arXiv [cs.CV]. arXiv. <http://arxiv.org/abs/2010.00747>.

[3] Chen, T., Kornblith, S., Swersky, K., Norouzi, M., & Hinton, G. (2020). Big self-supervised models are strong semi-supervised learners. arXiv preprint arXiv:2006.10029.

Some remarks about methodology/experimental section.

CheXpert dataset was used to determine condition-specific thresholds. However, then again, CheXpert was used for evaluation. Is this the correct approach? What about possible positive bias? How were the 500 images from cheXpert selected? Are these subset of images used for training?

Response:

We note that the CheXpert test dataset was not used to determine condition-specific thresholds. We use the CheXpert Validation dataset to determine condition-specific thresholds. When determining condition-specific thresholds to compute metrics such as F1 and MCC, we select thresholds that optimize Youden's index over the CheXpert *Validation* dataset which has no overlap with the CheXpert test dataset that is used for evaluation. We have clarified this in the *Evaluation* sub-section of the *Methods*.

*“The CheXpert **validation** dataset is utilized for tuning condition specific probability thresholds to transform the self-supervised model's probabilities for 14 different conditions of a given chest x-ray image into predictions. We conduct this analysis by running inference with the self-supervised model to obtain probability values of each condition being present for all chest x-ray images. Condition specific probability thresholds are then determined by choosing the probability values that result in the best Youden's J Statistic for each condition on the CheXpert validation dataset. The CheXpert validation dataset has no overlap with the CheXpert test dataset used for evaluation.”*

Furthermore, all 500 images used for evaluation are from the full CheXpert test dataset. This is a standard evaluation dataset that is used in the CheXpert competition [1]. We have also clarified this in the description of our *Evaluation* procedure.

“The CheXpert test dataset is a collection of chest x-rays that are commonly used to evaluate the performance of models on chest x-ray interpretation tasks^{14, 46}. We evaluate our model on the entire CheXpert test dataset, consisting of 500 chest x-ray images labeled for the presence of 14 different conditions⁸.”

[1] Irvin, Jeremy, Pranav Rajpurkar, Michael Ko, Yifan Yu, Silvana Ciurea-Ilicus, Chris Chute, Henrik Marklund, et al. 2019. “CheXpert: A Large Chest Radiograph Dataset with Uncertainty Labels and Expert Comparison.” In Proceedings of the AAAI Conference on Artificial Intelligence, 33:590–97.

The second dataset used for evaluation was PadChest. Again, the authors selected only 2978 images. Why? Why not use all 27% of images that have labels available.

Response:

The reviewer makes a great point. As per the suggestion of the reviewer, we have now rerun our PadChest evaluation procedure on all 27% of images (n = 39,053) that had human annotated labels available. We have updated Figure 3 and results pertaining to PadChest in the manuscript to reflect the results of the new experiment. We have included the updated Fig. 3 below for the reviewers' reference.

Fig. 3 | Performance on unseen radiographic findings in the `_PadChest_dataset`

Mean AUC and 95% CI is shown for each radiographic finding ($n > 50$) labeled as high importance by an expert radiologist. We externally validate the model's ability to generalize to different data distributions by evaluating model performance on the human-annotated subset of the PadChest dataset ($n = 39,053$ chest x-rays). No labeled samples were seen during training for any of the radiographic findings in this dataset. The self-supervised method achieves an AUC of at least 0.900 on 6 findings and at least 0.700 on 37 findings out of 57 radiographic findings where $n > 50$ in the PadChest test dataset ($n = 39,053$).

It would be interesting to compare with some of the recently proposed state-of-the art (self-supervised) approaches, not only comparison with the baseline model.

Response:

We have now added further self-supervised approaches as additional comparisons in Table 2.

Table 2 | Comparing the self-supervised method to supervised and self-supervised baselines on the CheXpert data set. Percentages refer to percentage of labels used in the training data. The self-supervised method nearly matches the supervised baseline is only -0.038 points below the highest performing fully supervised model on the CheXpert competition, Deep AUC Maximization (DAM)⁴⁶. DAN and outperforms the selfmi-supervised baselines ConVIRT, MedAug¹⁹, and MoCo-CXR¹⁸. Note the mean is over the 5 selected clinically relevant pathologies in the CheXpert data set.

	Model	Mean AUC
Supervised	DAM	0.931
	DenseNet-121	0.902
Self-Supervised	ConVIRT - 1%	0.870
	ConVIRT - 10%	0.881
	ConVIRT - 100%	0.881
	MedAug- 1%	0.810
	MoCo-CXR- 1%	0.802
	MoCo-CXR- 10%	0.850
	MoCo-CXR- 100%	0.884
	Ours - 0%	0.893

We have also modified the Results section to include:

“The self-supervised method’s mean AUC of 0.893 outperforms ConVIRT trained on 1% of labeled data (0.870 AUC), ConVIRT trained on 10% of labeled data (0.881 AUC), and ConVIRT trained 100% of labeled data (0.881 AUC), MedAug trained on 1% of labeled data (0.810), MoCo-CXR trained on 1% of labeled data (0.802 AUC), MoCo-CXR trained on 10% of labeled data (0.850 AUC), and MoCo-CXR trained on 100% of labeled data (0.884 AUC).”

Hints:

Table 3 is low quality, hard to read.

Response:

We assume the reader was referring to the quality of Figure 3, as we don’t have any tables in our main text (only in the supplement). We have worked to increase the quality of Figure 3 in our updated manuscript by substantially increasing the font size of all pathologies on the y-axis.

Is it necessary to define AUROC and MCC? These are notoriously known.

Response:

We have removed the definitions of AUROC and MCC as per the recommendation of the reviewer.

Some brief overview of similar works can be beneficial to the reader

Response:

We agree with the reviewer’s recommendation and have now added a brief overview of similar works:

“Our results show that our self-supervised method outperforms three previous label efficient methods (MoCo-CXR, MedAug, and ConVIRT) on the CheXpert dataset, while being the first

approach using no explicit labels during training. MoCo-CXR and MedAug utilize self-supervision using only chest x-ray images. Specifically, MoCo-CXR modifies the contrastive learning framework Momentum Contrast (MoCo) for chest x-ray interpretation. MedAug builds on MoCo pretraining by using patient metadata to select positive chest x-ray image pairs for the image-image contrastive pretraining. One prior work, ConVIRT, uses chest x-rays along with associated report data to conduct self-supervision. Specifically, ConVIRT jointly trains a ResNet50 and Transformer by leveraging randomly sampled text from paired chest x-ray and radiology report data to learn visual representations.”

Reviewer #2 (Report for the authors (Required)):

The proposed approach can match radiologists’ performance on multi-label pathology classification of chest X-rays and can generalize to pathologies that were not explicitly annotated for training. Additionally, the approach outperforms a fully-supervised model on three out of eight pathologies on an external validation set obtained from a hospital in a different country.

Overall, this is a paper demonstrating the potential of a previously published machine learning method (CLIP) [1] in pathology classification of chest x-rays. My concerns mainly come from its limited technical novelty (i.e., minor modifications on top of CLIP) and its design of the prompt engineering method. I consider the degree of advance of this work to be translational.

[1] Radford et al. Learning Transferable Visual Models From Natural Language Supervision. Proceedings of the 38th International Conference on Machine Learning, PMLR 139:8748-8763.

Response: We thank the reviewer for the summary of our work and suggestions, which we have addressed below.

I have the following concerns and comments.

1. One of my major concerns is that the proposed method seemed to be largely built upon CLIP (Contrastive Language-Image Pre-training) [1] (I also suggest to replace reference 15 with a more formal citation as CLIP has been published at ICML 2021). As far as I am concerned, the only obvious modification is that the authors replaced ResNet used in CLIP with ViT-B/32. Also, the authors did not give clarifications about the differences between the proposed approach and CLIP, which makes me feel the technical contribution of the proposed method is quite limited.

Response:

We have updated our citation for reference 15 as per the reviewer’s recommendation.

In response to the reviewer’s comment on technical differences from CLIP, we note that our proposed method draws its methodological novelty from building upon two independent works. In particular, we build upon the use of image-text pairings of chest x-rays and radiology reports in ConVIRT [1], as well as the multi-class zero-shot classification of natural images in CLIP [2]. As a result, our technical contributions help fill in the gaps left by prior works with regards to the

application of zero-shot approaches to medical image interpretation. In the following section, we aim to outline precisely our technical and methodological contributions to the field building upon these works. Additionally, for each subpoint, we include an excerpt from our manuscript contrasting our method with both ConVIRT and CLIP.

- *Enabling multi-label classification.* In [2], CLIP only demonstrates the ability to predict a single class for each image by outputting softmax normalized probability scores for each class. In doing so, their method normalizes across all possible classes, meaning a higher probability of one class implies lower probabilities of other classes. However, in medical images such as chest x-rays, there are often multiple pathologies or diagnoses which we would like to independently classify and obtain probabilities for. Our formulation of positive and negative pairs addresses this issue by normalizing each pathology with respect to its negation. As a result, we are able to make a prediction for each pathology independently.
 - *“Our proposed procedure allows us to normalize with respect to the negated version of the same disease classification instead of naively normalizing across the diseases to obtain probabilities from the logits, which is the original method proposed by CLIP¹⁵.”*
- *Careful selection of sections in text reports for training.* During image-text pair training, ConVIRT selects a random sentence from the full length radiology report for each image. Although this could extract some signal, a random text input selection allows for unnecessary stochasticity which could lead to inconsistencies in training. To address this, we select the text from the *Impressions* section.
 - *“We utilize the impressions section of each text report, since it contains a concise summary of the entire report. We contrast this with a prior self-supervised method, ConVIRT, which selects a random sentence from the full-length radiology report for each image¹⁴. Although their proposed method could extract some signal, a random text input selection allows for unnecessary stochasticity which could lead to inconsistencies in training. To address this, we consistently select the text from the impressions section.”*
- *Knowledge distillation procedure for longer text reports.* In [2], CLIP has a fixed context length for their text encoder, limiting the amount of textual information that it can learn from. Thus, only certain sections of corresponding text reports would have to be selected for training. However, full length medical reports often contain context which could act as a useful supervisory signal for medical image interpretation tasks. To address this limitation, we develop a knowledge distillation procedure which allows our method to scale to architectures with larger maximum token length (i.e. 512, which encompasses 98% of full length radiology reports in the MIMIC-CXR dataset) while using pre-trained weights from the best performing model with the CLIP default token length of 77.
 - *“To allow for the use of the CLIP pretrained model on full radiology reports, we use a knowledge distillation procedure. This procedure is required as the pretrained text encoder from the CLIP model has a context length of only 77 tokens, which is not long enough for an entire radiology report. We use the*

pretrained model to train a model with a context length of 512, long enough to encompass 98% of radiology reports.”

- *Training model pre-trained on natural images.* Additionally, we demonstrated that we can leverage the pre-trained weights from the CLIP architecture learned from natural images to train a zero-shot model with a domain-specific medical task.

[1] Zhang, Yuhao, Hang Jiang, Yasuhide Miura, Christopher D. Manning, and Curtis P. Langlotz. 2020. “Contrastive Learning of Medical Visual Representations from Paired Images and Text.” arXiv [cs.CV]. arXiv. <http://arxiv.org/abs/2010.00747>.

[2] Radford et al. Learning Transferable Visual Models From Natural Language Supervision. Proceedings of the 38th International Conference on Machine Learning, PMLR 139:8748-8763.

2. (Mandatory) Another major concern is that the proposed prompt engineering procedure in the Methods section seems to have serious flaws. The main steps in this procedure include 1) “A board-certified radiologist developed a list of 73 alternative prompts.” 2) “We then used the standard template of “__” and “not __” with each of the alternative labels and had the model predict the probability of the image corresponding to each label. Then, we were able to determine the best performing label for each pathology by looking at the AUC for each.” Here in the second step, computing AUC requires groundtruth pathology labels. It is not clear to me whether this prompt engineering procedure uses the CheXpert test dataset or a validation subset of the CheXpert training set. If the test set had been used, the authors would have used the groundtruth pathology labels of the test set to choose the best performing prompts, which should not be allowed. If a validation subset of the training set had been used, the authors would have used the groundtruth labels of the validation subset, which means the proposed method is “supervised” and the claims about zero-shot learning do not hold any more. Therefore, the proposed prompt engineering procedure is quite problematic either way.

Response:

We have clarified our writing and reported more experimental results to alleviate the reviewer’s concern.

We note that as a default, we run experiments using the labels present in the test set as the prompts, and creating the prompts of “<label>” and “not <label>” as the positive and negative prompts for the softmax evaluation procedure. Using these prompts requires no supervision as they are simply the labels from the test set. Our results on PadChest already use the default prompts.

We have also now added results on the CheXpert test set without the use of any alternative prompts, and included the results in Supplementary Table 8. As seen in the table, our performance remains unchanged for all CheXpert competition pathologies — on which our claims of coming close to the performance of radiologists and supervised models are based — except for Cardiomegaly, where the alternate prompt provides a boost in performance of +0.051. Nevertheless, our experiments with the alternative prompts serve to highlight the

potential for prompt engineering as a future avenue for exploration to improve the performance of our method even further.

Table 8 | Performance of the self-supervised method using default prompts on the CheXpert test data set

	Atelectasis	Cardiomegaly	Consolidation	Edema	Pleural Effusion
Default Prompts	0.844	0.844	0.871	0.875	0.915
Alternative Prompts	0.844	0.895	0.871	0.875	0.915

3. (Mandatory) Furthermore, the proposed method requires much effort from a board-certified radiologist to develop alternative prompts for every pathology. Although this effort does not directly assign pathology labels to individual x-ray images, it is a manual process and requires the domain knowledge of a human expert. Therefore, calling the proposed method “self-supervised” is not very rigorous. In addition, radiology reports were also originally produced by human experts although for a different purpose. I believe it would be less misleading if the authors can use terms like text-supervised or report-supervised learning to distinguish text-level supervision from image-level supervision.

Response:

As mentioned in CLIP [1], the terminology to describe learning image representations from unstructured text has been varied including unsupervised [2] weakly supervised [3] and self-supervised [4].

We have updated our introduction to clarify that our method is self-supervised, and uses natural language supervision with the following addition:

“Our method, which we call CheXzero, uses contrastive learning, a type of self-supervised learning, with image-text pairs to learn a representation that enables zero-shot classification. Our method can also be considered as a form of natural language supervision or unsupervised learning¹⁵.

[1] Radford et al. Learning Transferable Visual Models From Natural Language Supervision. Proceedings of the 38th International Conference on Machine Learning, PMLR 139:8748-8763.

[2] Zhang, Yuhao, Hang Jiang, Yasuhide Miura, Christopher D. Manning, and Curtis P. Langlotz. 2020. “Contrastive Learning of Medical Visual Representations from Paired Images and Text.” *arXiv Preprint arXiv:2010.00747*.

[3] Joulin, Armand, Laurens Van Der Maaten, Allan Jabri, and Nicolas Vasilache. "Learning visual features from large weakly supervised data." In *European Conference on Computer Vision*, pp. 67-84. Springer, Cham, 2016.

[4] L. Gomez, Y. Patel, M. Rusiñol, D. Karatzas and C. V. Jawahar, "Self-Supervised Learning of Visual Features through Embedding Images into Text Topic Spaces," *2017 IEEE Conference on Computer Vision and Pattern Recognition (CVPR)*, 2017, pp. 2017-2026, doi: 10.1109/CVPR.2017.218.

4. (Mandatory) Why did you need to “acquire free-text radiology reports corresponding to each of the 500 chest x-ray images to enable zero-shot evaluation”? These are the radiology reports of the CheXpert test dataset, and they have information related to the groundtruth pathology labels. Exploiting such information during performance evaluation on the test set should not be allowed.

Response:

We thank the reviewer for catching this; this was an error in our writing. We *did not* acquire free-text reports for each of the 500 chest x-ray images. We have clarified this in the updated manuscript by removing this sentence. Rather, we use the available labels in the CheXpert test dataset as ground-truth to evaluate the performance of our zero-shot predictions.

5. (Mandatory) All figures in the paper have low resolution, making it difficult to see more details.

Response:

The figures in the paper have been updated to have higher resolution. Additionally, we have created a folder with high resolution figures for easier analysis.

6. (Mandatory) Legends are missing in Fig. 2c. What do those points with 3 different colors stand for? Do they represent 3 radiologists? Then, why do their performance have so large gaps?

Response:

We have updated Fig. 2c to include legends for the AUC graphs, clarifying that each point represents the performance of one of three radiologists on the CheXpert test set.

We also assure the reviewer and have additionally clarified in the manuscript that the radiologist predictions used in our analysis are valid and have been verified by multiple rounds of reviewers as part of experiments conducted with CheXNeXt [1]. Thus, large gaps in radiologist performance are likely a result of human error.

[1] Rajpurkar, P., Irvin, J., Ball, R. L., Zhu, K., Yang, B., Mehta, H., ... & Lungren, M. P. (2018). Deep learning for chest radiograph diagnosis: A retrospective comparison of the CheXNeXt algorithm to practicing radiologists. *PLoS medicine*, 15(11), e1002686.

7. Comma is missing in the caption of Fig. 3 (2978->2,978).

Response:

We have updated these typos in the manuscript.

8. (Mandatory) In experimental results, what does DAN in the caption of Table 2 stand for? A reference should be added here. In Table 2, the authors used DAM. Do DAN and DAM have the same meaning or not?

Response:

This was a typo in the manuscript. We have changed the caption to DAM in the manuscript and have added a citation. We have also clarified the meaning of DAM by referencing the full title, Deep AUC Maximization, before using the abbreviation.

Reviewer #3 (Report for the authors (Required)):

The aim of this paper is to develop a SSL based method without explicit annotation labels to outperform a fully supervised learning with explicit annotation labels, which can match radiologists's performance on multi-label pathology classification with CXR. This is quite interesting topics. However, there are several concerns on the reproducibility of this study as follows.

1. The size of CXR (320x320) is too small to find lung nodule, which should be read in CXR to catch There is a lack of sample dataset. The size of CXR should be 512 x 512 or 1K x 1K to find the nodule in CXR.

Response:

We acknowledge this limitation of using 320x320 as an image size for finding lung nodules. The detection of lung nodules and other pathologies may certainly benefit from using a larger CXR image size. We decided to use the 320x320 resolution because it has been used by other popular fully-supervised methods, and have added the following line in the manuscript clarifying this choice.

“The image resolution used is the same as CheXpert⁸ and CheXNet¹, which both achieved radiologist level performance on external test sets.”

2. For radiologists' reading study, what is the size of CXR?

Response:

The radiologists were given full resolution images for reading. We have now clarified this with the following addition:

“Additionally, the test set contains predictions from 3 board certified radiologists on full resolution images to which we compare the performance of the model.”

3. Why do you select 5 pathologies in the CheXpert. In addition, other classes from 14 pathologies in the CheXpert should be declared.

Response:

We select these pathologies as they are used in the CheXpert competition. We add the following explanation to the datasets section:

“Additionally, we select the five pathologies of Atelectasis, Cardiomegaly, Consolidation, Edema and Pleural Effusion for comparison against other methods as they are the pathologies used in the CheXpert competition⁸”

Additionally, we have now specified the 14 pathologies in the data sets training section, with the line:

“The dataset is labeled for the presence of 14 different conditions: Atelectasis, Cardiomegaly, Consolidation, Edema, Enlarged Cardiomeastinum, Fracture, Lung Lesion, Lung Opacity, No Finding, Pleural Effusion, Pleural Other, Pneumonia, Pneumothorax, Support Devices”

4. The authors insist that their method outperform the supervised learning. However, there was only comparison to semi-supervised learning in Table 2, and 3. In this table, Mean AUC of supervised learning (0.931) is better than this method (0.915).

Response:

We would like to clarify that we do not state our method outperforms supervised learning. To further clarify this now, we have now modified our claim by specifying a numerical difference: *"our zero-shot method closely matches the performance of both expert radiologists and fully supervised methods on pathologies that were not explicitly labeled during training. Specifically, our method is -0.038 AUC points below the highest performing fully supervised model on the CheXpert competition."*

We have now added another comparison that utilizes the same backbone as self-supervised approaches but is trained in a supervised fashion such as a DenseNet-121. In table 2 we now thus have two comparisons to supervised approaches, DAM and a DenseNet-121 baseline. Deep AUC Maximization (DAM) uses a two step sophisticated supervised learning technique which requires the utilization of many datasets. Specifically, the CheXpert (224,316), Melanoma (46,131), DDSM+ (55,000), and PatchCamelyon (148,960) datasets are used for training. Additionally, DAM proposes a new AUC margin loss function to directly optimize for which allows for them to observe high results on the CheXpert dataset. The additional supervised comparison in Table 2 is of a DenseNet-121 trained on the CheXpert dataset that achieves a mean AUC of 0.902. Note that our method, in contrast, is only trained on the MIMIC-CXR dataset (377,110) with no labels and tested on the external CheXpert dataset and achieves a mean AUC of 0.893, close to the performance of the DenseNet-121.

5. There is a lack of details on this paper to authors considering the differences of semi-supervised methods in table 2 and 3, and other classes' result in table 3.

Response:

We have now added additional details about the differences between the method:

"Our results show that our self-supervised method outperforms three previous label efficient methods (MoCo-CXR, MedAug, and ConVIRT) on the CheXpert dataset, while being the first approach using no explicit labels during training. MoCo-CXR and MedAug utilize self-supervision using only chest x-ray images. Specifically, MoCo-CXR modifies the contrastive learning framework Momentum Contrast (MoCo) for chest x-ray interpretation. MedAug builds on MoCo pretraining by using patient metadata to select positive chest x-ray image pairs for the image-image contrastive pretraining. One prior work, ConVIRT, uses chest x-rays along with associated report data to conduct self-supervision. Specifically, ConVIRT jointly trains a ResNet50 and Transformer by leveraging randomly sampled text from paired chest x-ray and radiology report data to learn visual representations. "

Additionally, we have added additional experiments to compare the performance of the methods on all classes in table 2.

Rebuttal 2

Review Responses

Reviewer #1:

The authors answered/explained in the manuscript most of my remarks.

Response:

We thank the reviewer for their comments that improved our work, and have addressed the remaining remarks below.

I would like again raise the issue of novelty, especially in the context of some previous works such as:

1. Smit, Akshay, Saahil Jain, Pranav Rajpurkar, Anuj Pareek, Andrew Y. Ng, and Matthew P. Lungren. 436 2020. —CheXbert: Combining Automatic Labelers and Expert Annotations for Accurate Radiology

Report

437 Labeling Using BERT.

2. Vu, Yen Nhi Truong, Richard Wang, Niranjana Balachandrar, Can Liu, Andrew Y. Ng, and

455 Pranav Rajpurkar. 2021. —MedAug: Contrastive Learning Leveraging Patient Metadata Improves

456 Representations for Chest X-Ray Interpretation.

-The author explained novelty in general but not in the context of the previous published works of some of the co-authors.

Response:

We have clarified our novelty in relation to the previous works. In particular, with respect to [1], we detail that the system described in [1] still requires the existence of an automated labeler, and still relies on expert annotations on which the system is fine-tuned. We highlight this limitation in the discussion section: “the development time of automatic labeling systems such as the NIH labeler and CheXbert [1] is very high, each requiring either extensive domain knowledge or technical expertise to implement^{7,25}”. In contrast, the method we proposed here “is able to classify pathologies without requiring the domain-specific development of an automatic labeler.”

With respect to [2], we note that the system described in [2] requires a fine-tuning step to be able to predict the diseases using labeled data. We clarify this distinction in our discussion section: “*While self-supervised pre-training approaches have been demonstrated to increase label efficiency across several medical tasks, they still require a supervised fine-tuning step after pre-training that requires manually labeled data for the model to predict relevant pathologies*^{13, 14.}” (¹³ is MedAug and ¹⁴ is ConVIRT).

Another issue that I am not completely satisfied with is the review of the recent works. The authors add some recent papers, but mostly arxiv published papers. It would be better if the authors included preferably *peer-reviewed papers*.

Response:

We have now corrected our citations to reflect that these were in fact peer-reviewed papers, not ArXiv preprints. Our review of recent work includes MedAug in Proceedings of Machine Learning Research (PMLR) and CheXbert in the Conference on Empirical Methods in Natural Language Processing (EMNLP) 2020. We have replaced our original ArXiv citations for these works in our manuscript to highlight that they are peer-reviewed works.

Reference 13 and 19 in manuscript is the same one.

Response:

We have fixed this in the manuscript by removing the duplicate reference.

Otherwise, this is very solid work and an interesting application of ML.

Response:

We thank the reviewer for their helpful comments that have improved our manuscript.

Reviewer #2 (Report for the authors (Required)):

Thanks for the authors' detailed responses. However, two major concerns still remain and I have an additional concern about the inconsistency of the reported experimental results.

Response:

We thank the reviewer for their thoughtful suggestions, and have addressed the remaining concerns below.

1. The authors' responses as well as lines 269-274 and lines 327-329 in the revised manuscript confirmed that the validation set of CheXpert had been used for tuning hyperparameters, including probability thresholds. I would like to point out that the validation set is actually a part of training data, and hyperparameter tuning needs to access the labels of the validation set. This is contradictory to the most important claim that the proposed method is a zero-shot method, which is not supposed to touch any labels in the training data at all.

Response:

We note that the use of the validation set for hyperparameter tuning is a part of other well-recognized zero-shot methods. CLIP [1], one of the most well-known recent advances in zero-shot learning, recognizes this, writing, *"we repeatedly queried performance on full validation sets to guide the development of CLIP. Creating a new benchmark of tasks designed explicitly to evaluate broad zero-shot transfer capabilities, rather than re-using existing supervised datasets, would help address these issues."*

We thus maintain that our proposed method is a zero-shot method given this prior work, and have added a line to the limitations similarly acknowledging this: "The self-supervised method still requires repeatedly querying performance on a labeled validation set for hyperparameter selection and to determine condition specific probability thresholds when calculating MCC and F1 statistics."

[1] Radford et al. Learning Transferable Visual Models From Natural Language Supervision. Proceedings of the 38th International Conference on Machine Learning, PMLR 139:8748-8763.

2. The authors' responses and the captions of Tables 6 and 8 also confirmed that the majority of experimental results (including the results in Tables 1 and 2) on the CheXpert test set were obtained using best performing alternative prompts evaluated on the CheXpert test set itself. In particular, Table 1 shows the main results of the proposed CheXzero approach. Only Tables 7 and 8 report results obtained using default prompts. Note that using the test set to select best performing prompts is unacceptable even as a future avenue for exploration.

Response:

We now address this concern by **reporting all our results without the use of alternative prompts**. We have updated all results, including the MCC and F1 scores, to reflect the model's performance using default prompts of "{Condition}" vs. "No {Condition}". In order to achieve similar results to our model with alternative prompts, we report performance obtained using an ensemble of 10 models (what we

consider “the self-supervised model” throughout our paper). We summarize our main results in the *Results* section of our paper:

“On the Matthews correlation coefficient metric (MCC), there is no statistically significant difference (model – radiologist performance = -0.005 [95% CI -0.043, 0.034]) between the performance of the model (0.523 [95% CI 0.486, 0.561]) and that of the radiologists (0.530 [95% CI 0.499, 0.558]) averaged over the pathologies. On individual pathologies, the model’s MCC performance is higher, but not statistically significantly, compared to radiologists on Consolidation (0.018 [95% CI -0.090, 0.123]), Cardiomegaly (0.058 [95% CI -0.016, 0.133]), and Edema (0.015 [95% CI -0.070, 0.099]). The model’s MCC performance is lower, but not statistically significantly, compared to radiologists on Atelectasis (-0.078 [95% CI -0.154, 0.000]) and Pleural Effusion (-0.040 [95% CI -0.096, 0.013]). On the F1 metric, there is similarly no statistically significant difference (model – radiologist performance = -0.009 [95% CI -0.038, 0.018]) between the mean F1 performance of the model (0.606 [95% CI 0.571, 0.638]) and that of the radiologists (0.619 [95% CI 0.585, 0.642]) averaged over the pathologies. On individual pathologies, we find that the model F1 performance is significantly higher than that of radiologists on Cardiomegaly (model – radiologist performance = 0.065 [95% CI 0.013, 0.115]). We find that the model’s F1 performance is significantly lower than that of radiologists on Atelectasis (model – radiologist performance = -0.045 [95% CI -0.090, -0.001]). There are no statistically significant differences in F1 for Consolidation (model – radiologist performance = -0.050 [95% CI -0.146, 0.036]), Edema (model – radiologist performance = 0.018 [95% CI -0.053, 0.086]), and Pleural Effusion (model – radiologist performance = -0.034 [95% CI -0.078, 0.008]).”

We have updated Figure 2 to show these new results, which now displays results on the 5 CheXpert competition pathologies for consistency. We also include our updated results below.

Table 4 | Comparison of MCC’s of the self-supervised method and 3 board certified radiologists with 95% confidence intervals.

MCC	Average	Pleural Effusion	Edema	Atelectasis	Consolidation	Cardiomegaly
Radiologists (Mean)	0.530	0.671 (0.618, 0.727)	0.507 (0.431, 0.57)	0.548 (0.496, 0.606)	0.359 (0.262, 0.444)	0.566 (0.511, 0.620)
Ours	0.523	0.628 (0.558, 0.696)	0.520 (0.424, 0.616)	0.468 (0.396, 0.541)	0.374 (0.29, 0.458)	0.625 (0.553, 0.7)
Difference (Ours – Radiologist)	-0.005 (-0.043, 0.034)	-0.04 (-0.096, 0.013)	0.015 (-0.070, 0.099)	-0.078 (-0.154, 0.000)	0.018 (-0.090, 0.123)	0.058 (-0.016, 0.133)

Table 5: Comparison of F1’s of the self-supervised method and 3 board certified radiologists with 95% confidence intervals.

F1	Average	Pleural Effusion	Edema	Atelectasis	Consolidation	Cardiomegaly
Radiologists (Mean)	0.619	0.737 (0.689, 0.783)	0.583 (0.511, 0.645)	0.692 (0.646, 0.731)	0.385 (0.28, 0.485)	0.678 (0.634, 0.718)

Ours	0.606	0.704 (0.634, 0.764)	0.602 (0.517, 0.678)	0.646 (0.593, 0.700)	0.333 (0.239, 0.424)	0.743 (0.685, 0.793)
Difference (Ours – Radiologist)	-0.009 (-0.038, 0.018)	-0.034 (-0.078, 0.008)	0.018 (-0.053, 0.086)	-0.045 (-0.090, -0.001)	-0.05 (-0.146, 0.036)	0.065 (0.013, 0.115)

3. The results reported in Tables 6 and 8 are inconsistent. As stated by the authors, the results in Tables 6 and 8 are reported on the CheXpert test set. Thus, the improvements achieved with the best performing alternative prompts over the default prompts should be the same in both tables. However, I noticed that at least half of them are not consistent. For example, over Consolidation and Edema, Table 6 shows that the best performing alternative prompts achieve 0.039 and 0.019 improvements in AUC, but in Table 8, the alternative prompts do not bring any improvements. In contrast, the results on Atelectasis and Cardiomegaly are consistent.

Response:

We have removed all usages of alternative prompts and discarded the associated tables, thereby addressing this concern.

Reviewer #3 (Report for the authors (Required)):

Thank for your kind response. However, the size of CXR could be critical for clinical application of your methods.

Response:

We have added a line in the *Discussion* to acknowledge this concern:

“Future work should develop approaches to scale this method to larger image sizes to better classify smaller pathologies.”

Rebuttal 3

Review Responses

My previous concerns have been addressed in this revised version. However, I have the following new concerns.

We thank the reviewer for appreciating our previous revisions. We have responded to the reviewer's new suggestions below.

1) The main results of the proposed method reported in this version was obtained using an ensemble of 10 models while the performance of those compared self-supervised methods was obtained from single models. Such comparisons are unfair. I notice the mean AUC of the best single model of the proposed method, as shown in Table 7, is 0.878, which is almost the same as the performance of ConVIRT (0.870) when 1% training data is used.

We believe that our comparisons are fair. The choice and method of ensembling is part of the model development itself, and it is standard for this to be considered a part of the proposed method: several methods on the CheXpert leaderboard build ensemble models for evaluation [1, 2]. We note that the best supervised method we compare to, DeepAUC [3], is also a method that similarly leverages an ensemble of models.

Even with the single model performance, consider that we are outperforming ConVIRT without the use of any training labels: we achieve an AUC of 0.878 with 0 training examples, compared to ConVIRT's AUC of 0.870 with 1000+ training examples. We note that ConVIRT is not a published method (the work is in preprint format), and we have demonstrated even stronger results over published methods, including MedAug - 1% (0.810 AUC), MoCo-CXR - 1% (0.802 AUC) and even MoCo-CXR - 10% (which uses 10,000+ training examples) (0.850 AUC).

[1] Pham, Hieu H., Tung T. Le, Dat Q. Tran, Dat T. Ngo, and Ha Q. Nguyen. "Interpreting chest X-rays via CNNs that exploit hierarchical disease dependencies and uncertainty labels." *Neurocomputing* 437 (2021): 186-194.

[2] Irvin, Jeremy, Pranav Rajpurkar, Michael Ko, Yifan Yu, Silvana Ciurea-Ilcus, Chris Chute, Henrik Marklund, et al. 2019. "Chexpert: A Large Chest Radiograph Dataset with Uncertainty Labels and Expert Comparison." In *Proceedings of the AAAI Conference on Artificial Intelligence*, 33:590–97.

[3] Yuan, Zhuoning, Yan Yan, Milan Sonka, and Tianbao Yang. 2020. "Robust Deep AUC Maximization: A New Surrogate Loss and Empirical Studies on Medical Image Classification." *arXiv Preprint arXiv:2012.03173*.

2) The proposed method uses ViT as the backbone while ConVIRT uses ResNet50 as the backbone for image feature extraction (Transformer is used for text reports not for images). This is also unfair as ViT is more powerful than ResNet50.

We note that ConVIRT does not leverage ViT but instead uses other tricks that we do not use at all. The ConVIRT method includes a ResNet50 model pre-trained on the ImageNet dataset; a

text model that uses the BERT architecture that is initialized with the ClinicalBERT model pre-trained on the MIMIC clinical notes. Further tricks used by ConVIRT include application of *“random cropping with a ratio sampled from [0.6, 1.0]; horizontal flipping with $p = 0.5$; affine transformation with a degree sampled from $[-20, 20]$, max horizontal and vertical translation fractions of 0.1, and a scaling factor sampled from [0.95, 1.05]; color jittering with brightness and contrast adjustment ratios sampled from [0.6, 1.4]; and Gaussian blur with $\sigma \in [0.1, 3.0]$.”*

The use of the ViT is an innovative characteristic of our method. By leveraging the architecture proposed in CLIP [1], we are able to leverage pre-trained weights from CLIP, which we then further train on a corpus of image and report pairs. This way, we are able to benefit from an initialization that already has learned some alignment between images and text.

Standardizing the backbone between ConVIRT and our method would thus not be a useful control: the choice of architecture is as much a core piece of the design of the method as is the loss function, the optimization and regularization strategy. We demonstrate that even without the biomedically-pretrained text encoder used by ConVIRT and the extra tricks for augmenting images using cropping/flipping/color-jittering/blurring, our method is able to achieve high performance.

[1] Radford et al. Learning Transferable Visual Models From Natural Language Supervision. Proceedings of the 38th International Conference on Machine Learning, PMLR 139:8748-8763.

I would like to see a comparison where both ConVIRT and the proposed method use the same backbone for image feature extraction and their performance is measured on single models. To warrant publication in NBME, the proposed zero-shot method should clearly outperform ConVIRT trained with 1% training data.

We have responded to both of the reviewer's suggestions above. We thank the reviewer once again for their thoughtful comments and suggestions throughout the review process!

Rebuttal 4

Review Responses

I have to disagree with the authors' latest responses.

1) Ensembling could be added to many machine learning algorithms for the purpose of performance boosting. If ensembling has to be used, for the sake of fairness, it should be added to all algorithms participating in a comparison.

We note that we have not relied on ensembling to make our argument. Importantly, **without ensembling, our method achieves an AUC of 0.878 with 0 training examples, compared to ConVIRT's AUC of 0.870 with 1000+ training examples.**

2) The authors wrote that ConVIRT includes a ResNet50 model pre-trained on the ImageNet dataset; a text model that uses the BERT architecture that is initialized with the ClinicalBERT model pre-trained on the MIMIC clinical notes; and some data augmentation strategies. I notice that the ImageNet dataset has around 1 million training images and the MIMIC-CXR dataset has 217k image-text pairs. However, the authors did not mention that the pretrained ViT model they took from CLIP was trained on 400 million image-text pairs collected from the internet. It is obvious that the authors' model has unfair advantages, including a more powerful backbone and a much larger pretraining dataset. If the proposed method cannot switch its backbone to ResNet50, in a fair comparison, ConVIRT should be revised to replace its backbone with the pretrained ViT model used in the current manuscript.

We have revised ConVIRT to replace its backbone with the pretrained ViT model used in the current manuscript as the reviewer suggests. **We find that even when using the ViT backbone, ConVIRT-ViT still underperforms our best single zero-shot model.**

We compare the mean AUC of ConVIRT-ViT using 1%, 10% and 100% label fractions to our single best model on the CheXpert validation dataset.

	Model	Mean AUC
	ConVIRT-ViT - 1%	0.725
	ConVIRT-ViT - 10%	0.809
	ConVIRT-ViT - 100%	0.856
	Ours - 0% (Best Single Model)	0.878

We obtained these results by using ConVIRT's codebase and replacing the ResNet-50 architecture with the pretrained ViT from CLIP, pre-training on MIMIC-CXR, and fine-tuning the model on the CheXpert dataset at 1%, 10% and 100% of labels using ConVIRT's scripts. To ensure a fair comparison, we performed a hyperparameter sweep on ConVIRT-ViT when fine-tuning; the best results from this process are presented above and in the manuscript.

Additionally, we note that the reported ConVIRT results are derived from *fine-tuning* the full backbone on the CheXpert training dataset. Despite ConVIRT's advantage, our *best single zero-shot model* is able to outperform ConVIRT's fine-tuned model without requiring any parameter updates to the backbone.

We have included this additional comparison in Table 3 of our manuscript comparing our method to prior self-supervised methods.

3) The proposed method should be compared with the following two state-of-the-art papers in a fair manner (the same backbone and the same setting regarding model ensembling), and should demonstrate clearly better performance. Although the first paper (ConVIRT) is still in preprint format, it was posted on arXiv about 18 months ago in October 2020 and has been widely regarded as a state-of-the-art method. The second paper is a published ICCV 2021 paper, but is not compared with in the current manuscript.

Yuhao Zhang, Hang Jiang, Yasuhide Miura, Christopher D Manning, and Curtis P Langlotz. Contrastive learning of medical visual representations from paired images and text. arXiv preprint arXiv:2010.00747, 2020.

Shih-Cheng Huang, Liyue Shen, Matthew P. Lungren, and Serena Yeung. GLoRIA: A Multimodal Global-Local Representation Learning Framework for Label-efficient Medical Image Recognition. ICCV 2021.

We have now included comparisons to both of these methods, and demonstrated that our single model outperforms both methods.

Comparisons to the first paper (ConVIRT) after standardizing the backbone and setting regarding model ensembling have been noted in our point above.

We now have additionally made a comparison with GLoRIA. In particular, we run GLoRIA's zero-shot method on the same multi-label CheXpert test dataset that we evaluate our method on. We find that GLoRIA obtains a Mean AUC of 0.534, demonstrating that GLoRIA is unable to generalize well to *multi-label classification on CheXpert*: in their work, they reported results using a single-label subset of the CheXpert test dataset). This highlights the known limitation of GLoRIA which is that the method only performs well when only a single pathology is present. Our previous version of manuscript had already included a discussion point surrounding this limitation of GLoRIA: "Recent work has leveraged radiology reports for zero-shot chest x-ray classification; however, it is only applicable to chest x-ray images with only one pathology,

limiting the practicality of the method since multiple pathologies are often present in real-world settings 23”

We have now added the GLoRIA comparison to Table 3 of our manuscript comparing our method to other self-supervised methods.

We report per-pathology results below, along with a direct per-pathology comparison to our best *single zero-shot model*.

	Mean AUC	Atelectasis	Cardiomegaly	Consolidation	Edema	Pleural Effusion
GLoRIA	0.534	0.339	0.553	0.485	0.655	0.569
Ours (Best Single Model)	0.878	0.816	0.881	0.904	0.888	0.927

We once again thank the reviewer for their deliberate review and consideration.